# Aberrant subchondral osteoblastic metabolism modifies Na$_V$1.8 for osteoarthritis

Jianxi Zhu[1,2], Gehua Zhen[1], Senbo An[1,2], Xiao Wang[1], Mei Wan[1], Yusheng Li[1,2], Zhiyong Chen[3], Yun Guan[3], Xinzhong Dong[4], Yihe Hu[2]*, Xu Cao[1]*

[1]Departments of Orthopaedic Surgery and Biomedical Engineering and Institute of Cell Engineering, The Johns Hopkins University School of Medicine, Baltimore, United States; [2]Department of Orthopaedic Surgery, Xiangya Hospital, Central South University, Changsha, China; [3]Department of Anesthesiology and Critical Care Medicine, The Johns Hopkins University School of Medicine, Baltimore, United States; [4]Department of Neuroscience, Neurosurgery, and Dermatology, Center of Sensory Biology, The Johns Hopkins University School of Medicine, Howard Hughes Medical Institute, Baltimore, United States

**Abstract** Pain is the most prominent symptom of osteoarthritis (OA) progression. However, the relationship between pain and OA progression remains largely unknown. Here we report osteoblast secret prostaglandin E2 (PGE2) during aberrant subchondral bone remodeling induces pain and OA progression in mice. Specific deletion of the major PGE2 producing enzyme cyclooxygenase 2 (COX2) in osteoblasts or PGE2 receptor EP4 in peripheral nerve markedly ameliorates OA symptoms. Mechanistically, PGE2 sensitizes dorsal root ganglia (DRG) neurons by modifying the voltage-gated sodium channel Na$_V$1.8, evidenced by that genetically or pharmacologically inhibiting Na$_V$1.8 in DRG neurons can substantially attenuate OA. Moreover, drugs targeting aberrant subchondral bone remodeling also attenuates OA through rebalancing PGE2 production and Na$_V$1.8 modification. Thus, aberrant subchondral remodeling induced Na$_V$1.8 neuronal modification is an important player in OA and is a potential therapeutic target in multiple skeletal degenerative diseases.

*For correspondence:
huyh1964@163.com (YH);
xcao11@jhmi.edu (XC)

**Competing interests:** The authors declare that no competing interests exist.

## Introduction

Subchondral bone is an integral component of the joint, absorbing compressive forces during movement (*Martel-Pelletier et al., 2016*). Physiological subchondral bone remodeling maintains its structural integrity and supports the overlying articular cartilage. Age or trauma (*Hayami et al., 2004*) related alteration of subchondral bone is a principal risk factor of osteoarthritis (OA), the most common joint disease (*Berenbaum et al., 2018*) characterized by cartilage destruction (*Glasson et al., 2005*; *Kim et al., 2014a*), subchondral sclerosis (*Suri et al., 2007*) and synovitis (*Benito et al., 2005*; *Sellam and Berenbaum, 2010*). Pain, as the major symptom for OA patients (*Lane et al., 2010*), often leads to physical disability and mortality in senior patients (*Chen et al., 2017*). Although the major local source is not clearly defined (*Malfait and Schnitzer, 2013*), there is evidence showing that synovitis (*Sellam and Berenbaum, 2010*) and subchondral bone marrow lesion (BML) are highly relevant to OA pain. BML is a fluid enriched area under magnetic resonance imaging (MRI) and is characterized as bone marrow edema, fibrosis, microfractures or trabecular pattern alterations in pathological examinations, with high relevance to abnormal bone remodeling. However, how subchondral BML induces OA pain, still remains largely unknown. We previously showed that aberrant subchondral bone remodeling in response to altered

 

**eLife digest** Many people will suffer from joint pain as they age, particularly in their knees. The most common cause of this pain is osteoarthritis, a disease that affects a tissue inside joints called cartilage. In a healthy knee, cartilage acts as a shock absorber. It cushions the ends of bones and enables them to move smoothly against one another. But in osteoarthritis, cartilage gradually wears away. As a result, the bones within a joint rub against each other whenever a person moves. This makes activities such as running or climbing stairs painful.

But how does this pain arise? Previous work has implicated cells called osteoblasts. Osteoblasts are found in the area of the bone just below the cartilage. They produce new bone tissue throughout our lives, enabling our bones to regenerate and repair. Each time we move, forces acting on the knee joint activate osteoblasts. The cells respond by releasing a key molecule called PGE2, which is a factor in pain pathways. The joints of people with osteoarthritis produce too much PGE2. But exactly how this leads to increased pain sensation has been unclear.

Zhu et al. now complete this story by working out how PGE2 triggers pain. Experiments in mice reveal that PGE2 irritates the nerve fibers that carry pain signals from the knee joint to the brain. It does this by activating a channel protein called $Na_v1.8$, which allows sodium ions through the membranes of those nerve fibers. Zhu et al. show that, in a mouse model of osteoarthritis, $Na_v1.8$ opens too widely in response to binding of PGE2, so the nerve cells become overactive and transmit a stronger pain sensation. This means that even small movements cause intense pain signals to travel from the joints to the brain.

Building on their findings, Zhu et al. developed a drug that acts directly on bone to reduce PGE2 production, and show that this drug reduces pain in mice with osteoarthritis. At present, there are no treatments that reverse the damage that occurs during osteoarthritis, but further testing will determine whether this new drug could one day relieve joint pain in patients.

mechanical loading patterns in OA was initiated by over-activation of transforming growth factor β1 (TGF-β1). High levels of subchondral TGF-β1 signaling induces mesenchymal stem cells (MSC) clustering and leads to the uncoupling of osteoblastic bone formation, osteoclastic bone resorption and angiogenesis. Moreover, we found overactivated osteoclasts in BML aggravated OA pain by secreting the axon guidance molecule Netrin-1 to induce subchondral sensory innervation (*Zhu et al., 2019*). The increased sensory innervation provides a structural base for the transmission of nociceptive signals from the subchondral bone to the central nervous system. However, how these nerve fibers are activated and sensitized during OA progression remains to be elucidated.

The persistent low grade of local inflammation is another hallmark of OA (*Midwood et al., 2009*; *Tu et al., 2019*). During the uncoupled bone remodeling in OA progression, a series of pro-inflammatory factors (*Kapoor et al., 2011*; *Sellam and Berenbaum, 2010*) like prostaglandin E2 (PGE2) (*Chen et al., 2019*), interleukin-1β (IL-1β), interleukin-6 (IL-6) are released into the subchondral bone area (*Massicotte et al., 2002*). We previously showed that PGE2 was a crucial factor in both pain sensation and bone metabolism (*Tu et al., 2019*). However, the main source remains largely unknown. We newly discovered a feedback mechanism on sensory nerve regulation of bone mass. PGE2 concentration is inversely related to bone mass and sensory nerves monitors bone density by responding to the concentration of PGE2 in bone (*Chen et al., 2019*). We found that when bone mass is decreased, the enzymatic activity of cyclooxygenase 2 (*Cox2*) in osteoblastic lineage cells was significantly increased and thus catalyzes more arachidonic acid into PGE2. At the early stage of OA, the bone density is temporarily decreased, which resembles the low bone density as seen in osteoporosis. Currently, there is still lack of information whether the increased PGE2 in OA subchondral bone is also attributed to the increased *Cox2* activity in osteoblastic cells in response to low bone mass.

PGE2 functions as an inflammatory mediator and a neuromodulator that alters neuronal excitability (*Samad et al., 2001*). In the four types of G-protein-coupled EP receptors (EP1-EP4) that mediate the functions of PGE2, EP4 receptor is considered as the primary mediator of PGE2- evoked inflammatory pain hypersensitivity and sensitization of sensory neurons (*Boyd et al., 2011*; *Chen et al., 2008*; *Lin et al., 2006*; *McCoy et al., 2002*; *Nakao et al., 2007*; *Southall and Vasko,*

*2001*; *Taylor-Clark et al., 2008*), as evidenced by that the specific EP4 receptor antagonists could reduce acute and chronic pain (*Nakao et al., 2007*), including OA pain (*Abdel-Magid, 2014*). Increased neuronal excitability contributes to the generation of hypersensitivity in various types of chronic pain (*Kuner, 2010*; *Malfait and Schnitzer, 2013*). PGE2 has been shown to potentiate several ion channels in neurons to enhance neuronal excitability (*Funk, 2001*). Voltage-gated sodium channel ($Na_V$), a member of the tetrodotoxin-resistant sodium channel (TTX-R), is mainly expressed in small- and medium-sized Dorsal root ganglion (DRG) neurons and their fibers. The $Na_V$ is responsible for initiating and propagating electrical signal transmission by inducing $Na^+$ influx to start action potential firing. PGE2 has been shown to modulate the sodium current of the TTX-R in DRG neurons and promote Nav1.8 trafficking to the cell surface (*England et al., 1996*; *Liu et al., 2010*). Therefore, the PGE2 induced neuronal hypersensitivity is likely to be mediated by the Nav 1.8 during OA progression.

Among the 9 subtypes of $Na_V$s ($Na_V$1.1–1.9), $Na_V$1.8 (*Akopian et al., 1996*) is the main drug target due to its highly relevant to pain signal transmission, and restricted distribution in primary nociceptive neurons (*Akopian et al., 1999*; *Julius and Basbaum, 2001*). Gain of function mutations in human in the promoter region of $Na_V$1.8 directly induces pain hypersensitivity (*Duan et al., 2018*). Interestingly, animals lacking $Na_V$1.8 display significant lower mechanical pain sensitivity with modest changes in heat or innocuous touch sensitivities (*Akopian et al., 1999*; *Basbaum et al., 2009*). This specificity of $Na_V$1.8 in transducing mechanical pain signals makes it highly possible in the participation of mechanical allodynia in OA. Moreover, post-transcriptional modifications of $Na_V$1.8 including phosphorylation (*Gold et al., 1998*; *Hudmon et al., 2008*; *Wu et al., 2012*) and methylglyoxalation (*Bierhaus et al., 2012*) further regulate its activity. A recent study demonstrated that inhibition of the expression of $Na_V$s in nociceptive neurons was effective in OA pain alleviation (*Miller et al., 2017*), with the detailed molecular mechanism remained to be clarified (*Strickland et al., 2008*).

In this study, we take the initiative to show that aberrant subchondral bone remodeling contributes to neuronal hypersensitivity during OA progression. Excessive PGE2 is synthesized by osteoblastic lineage cells in response to the low bone density at the early stage of OA. Excessive PGE2 sensitize sensory fibers innervates subchondral bone by upregulating the expression of sodium channel $Na_V$1.8 in both subchondral bone nerve fibers and DRG neuron body, which contributes to peripheral mechanical allodynia during OA progression. Therefore, we developed a small molecule conjugate by linking the TGFβ type receptor 1 (TβlR) inhibitor (LY-2109761) and alendronate (Aln) (*Hayami et al., 2004*) to achieve bone-targeted delivery. We used this conjugate (Aln-Ly) as a proof of concept drug to test whether reversing the aberrant bone remodeling by synergistically inhibiting osteoclast bone resorption and the excessive TGF-β activity can substantially reduce the PGE2 production and subsequent mechanical hypersensitivity that generated in OA subchondral bone.

## Results

### $Na_V$1.8 was modified in the subchondral bone and mediate OA progression

To identify the primary voltage-gated sodium channel in subchondral sensory fibers that responsible for mechanical hypersensitivity during OA progression, we tested the expression levels of different sensory related sodium channel $Na_V$s in OA mice post anterior cruciate ligament transection (ACLT). The transcription levels of mRNAs that encode $Na_V$s in DRG including $Na_V$1.1, $Na_V$1.2, $Na_V$1.3, $Na_V$1.6, $Na_V$1.7, $Na_V$1.8, $Na_V$1.9 were measured by qPCR using mRNA isolated from mouse ipsilateral L3-5 DRGs one month post-ACLT or sham surgery. Compares to the sham-operated group, the expression of mRNA encoding $Na_V$1.8 increased 2.5 folds in the ACLT group as the highest upregulation among all the $Na_V$s. The mRNAs encoding $Na_V$1.7 and $Na_V$1.9 showed moderate upregulation in the ACLT group relative to that of the Sham group while the changes of $Na_V$1.1 $Na_V$1.2 $Na_V$1.3 or $Na_V$1.6 were not detected (*Figure 1a*). Therefore, we further investigated $Na_V$1.8 protein expression in the immune-histological analysis of OA subchondral bone. The intensity of $Na_V$1.8 immunofluorescence in subchondral bone was also elevated about 2 to 3 folds in OA mice compared to sham-operated mice one- or two-months post-surgery (*Figure 1b and c*). To examine whether the increase

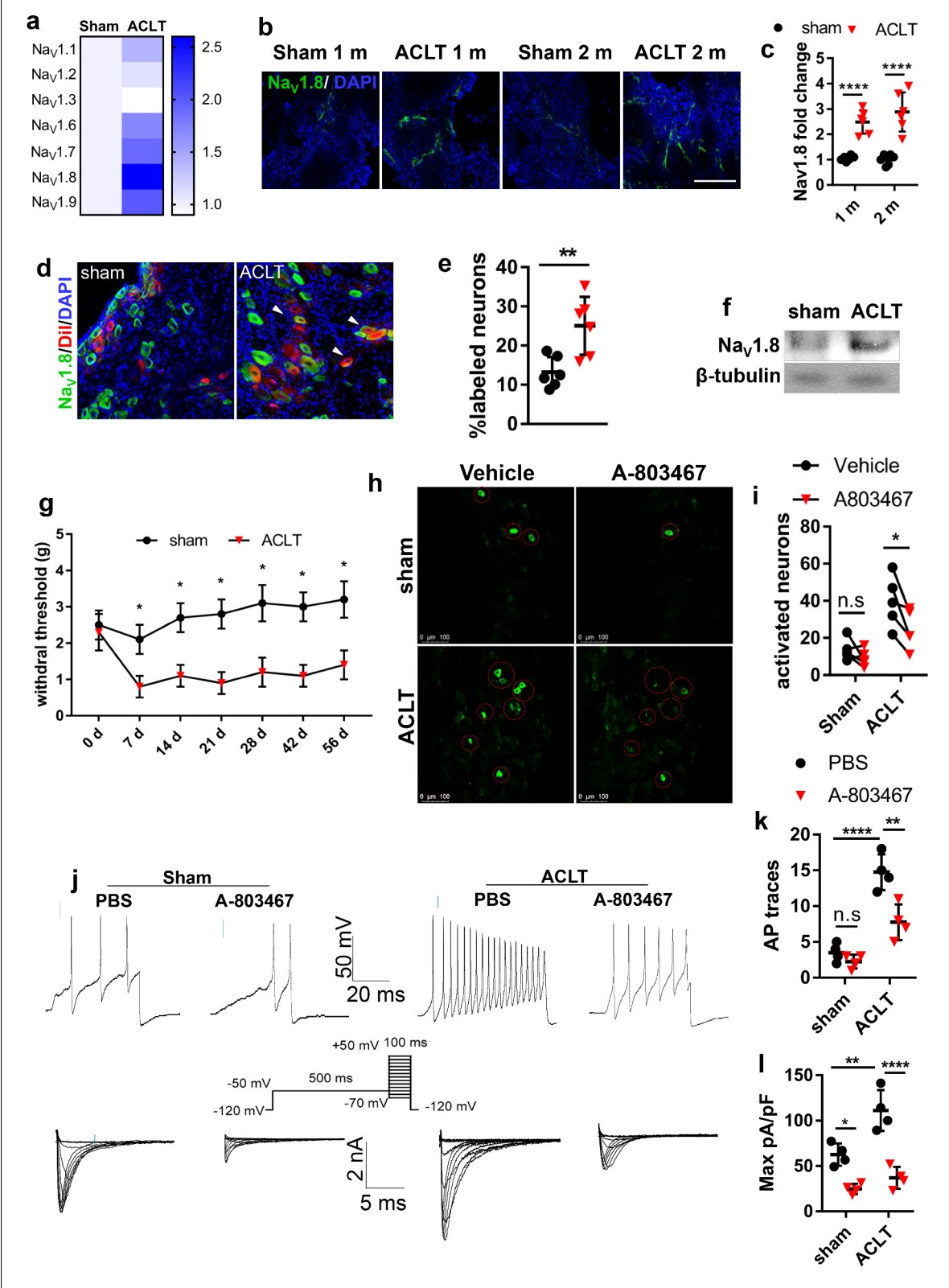

**Figure 1.** Nav1.8 modification after mice model of OA. (a) Heatmap of relative expression levels of Na$_V$ channels in ipsilateral L3-L5 DRGs after sham or ACLT surgery. (b, c) Immunostaining of Na$_V$1.8$^+$ (green) nerve fibers (b) and statistical analysis (c) in mouse tibial subchondral bone after sham or ACLT surgery at 1 m and 2 m. Scale bars, 20 μm, n = 6 per group. (d, e) Immunostaining of Na$_V$1.8$^+$ (green) nerve fibers (d) and statistical analysis (e) in ipsilateral sciatic nerve after sham or ACLT surgery at 1 m. Scale bars, 40 μm, n = 6 per group. (f) Western blots of Na$_V$1.8 in mouse ipsilateral L3-5

*Figure 1 continued on next page*

*Figure 1 continued*

DRGs 1 month post sham or ACLT surgery. the experiment was repeated three times and a representative result was chosen. (g, h) Retrograde tracing of Nav1.8 (green) and DiI (red) and DAPI (blue) double-labeled neurons (g) and percentage of double labeled neurons (h) in ipsilateral L4 DRG of rat after sham or ACLT surgery in 3 m. Scale bar, 80 μm. n = 6 per group. **p<0.01, ***p<0.001, ****p<0.0001 compared with the sham-operated group at different time points. Statistical significance was determined by multifactorial ANOVA WITH BONFERRONI POST HOC TEST (c, k, l), unpaired Student's *t* test (e an i) and all data are shown as scattered plots with means ± standard deviations. (h, i) Representative photomicrographs (h) and statistically analysis of activated neurons (i) in ipsilateral L4 DRG using in vivo Pirt-GCaMP3 imaging treated before or after A803467 1 month post sham or ACLT surgery. n = 6 per group. (j–l) Representative traces of Aps (j upper), maximal current density (j lower), statistical analysis of AP numbers (k) and Na$_V$1.8 currents (l) of DRG 1 month post sham or ACLT. **p<0.01, ***p<0.001, ****p<0.0001 compared with the sham-operated group at different time points. Statistical significance was determined by multifactorial ANOVA WITH BONFERRONI POST HOC TEST (c, k, l), unpaired Student's *t* test (e an i) and all data are shown as scattered plots with means ± standard deviations.

The online version of this article includes the following source data and figure supplement(s) for figure 1:

**Source data 1.** Raw data of Navs QPCR, subchondral Nav1.8 fiber density, Retrograde tracing, Von Frey tests, GcAMP3 imaging, and electrophysiological recordings.
**Source data 2.** Full scan of western blots in *Figure 1f*.
**Figure supplement 1.** Na$_V$1.8 upregulation and colocalization in different subsets of sensory neurons in vivo and in vitro.

of Na$_V$1.8 expression is limited to in a certain subtype(s) of the DRG neuron that innervates subchondral bone in OA mice, Na$_V$1.8 was co-stained with different markers for sensory nerve subtypes based on the current classification of DRG neurons (*Usoskin et al., 2015*). The expression rate of Na$_V$1.8 in total nerve fibers (labeled by pan neuron marker PGP9.5) innervated in subchondral bone significantly increased post-ACLT (*Figure 1—figure supplement 1a,f*). Moreover, the elevated Na$_V$1.8 expression was highly co-localized with the peptidergic nociceptor marked by calcitonin gene-related peptide (CGRP) (*Brain et al., 1985*; *Figure 1—figure supplement 1b,f*) and mechanosensitive low-threshold mechanoceptors (labeled by PIEZO2) (*Eijkelkamp et al., 2013*). The expression of Na$_V$1.8 was also slightly elevated in the synovium (*Figure 1—figure supplement 1f* 1 hr, j). Both western blot analysis (*Figure 1f*) and immunostaining (*Figure 1—figure supplement 1f* 1 g, i) of ipsilateral lumbar 3–5 DRG confirmed the upregulation of Na$_V$1.8 expression at the DRG level. We then further validated whether DRG neurons with upregulated Na$_V$1.8 expression directly innervates fibers in the subchondral bone. We injected a neurophilic fluorescent dye (DiI) into the subchondral bone to label the distal nerve fibers (*Ferreira-Gomes et al., 2010*). We found that the Na$_V$1.8$^+$ neurons labeled by DiI significantly increased in the DRG neurons of OA rats relative to sham-operated rats (*Figure 1g,h*), indicating that DiI was transported into DRG neurons by the sensory fibers innervated subchondral bone in a retrograde manner. These results suggest that the expression of pain-related sodium channel Na$_V$1.8 is upregulated in DRG neurons and their axons that innervate subchondral bone during progression. We then examined the association between Na$_V$1.8 neuronal activity and OA pain. Von Frey test showed that the hind paw withdrawal threshold (HPWT) dropped nearly 60% and maintained at this level throughout the two month-period post ACLT relative to the sham-operated group, suggesting that development of mechanical allodynia in OA mice (*Figure 1g*; *Chen et al., 2017*). To assess the potential role of Na$_V$1.8 in DRG neuronal excitability, we used an in vivo DRG imaging in Pirt$^{GCaMP3fl/-}$ mice that we recently developed. In this genetically targeted mice, genetic-encoded Ca$^{2+}$ indicator GCaMP3 is specifically expressed in >95% of all DRG neurons under the control of the Pirt promoter. In the Pirt$^{GCaMP3fl/-}$ mice, the excitability of the nociceptive neurons in DRG can be visualized by fluorescence signals of calcium influx. The number of excited DRG neurons ipsilateral to the surgery significantly increased in OA mice compared to sham-operated mice, and importantly, administration of Na$_V$1.8 inhibitor (A-803467)(*Jarvis et al., 2007*) blunted the signal in DRG (*Figure 1h and i*). To validate the excitation of DRG neurons related to Na$_V$1.8, we performed patch-clamp in the DRG neurons that were isolated from mice that underwent ACLT or sham surgery. The action potential number and Na$_V$1.8 current density significantly elevated in ACLT mice relative to sham-operated mice, and the elevations were blocked by A-803467 (*Figure 1j–l*, *Figure 1—figure supplement 1f* 1 k and l). Thus, the activation of Na$_V$1.8 mediates OA pain related DRG neuron hypersensitivity.

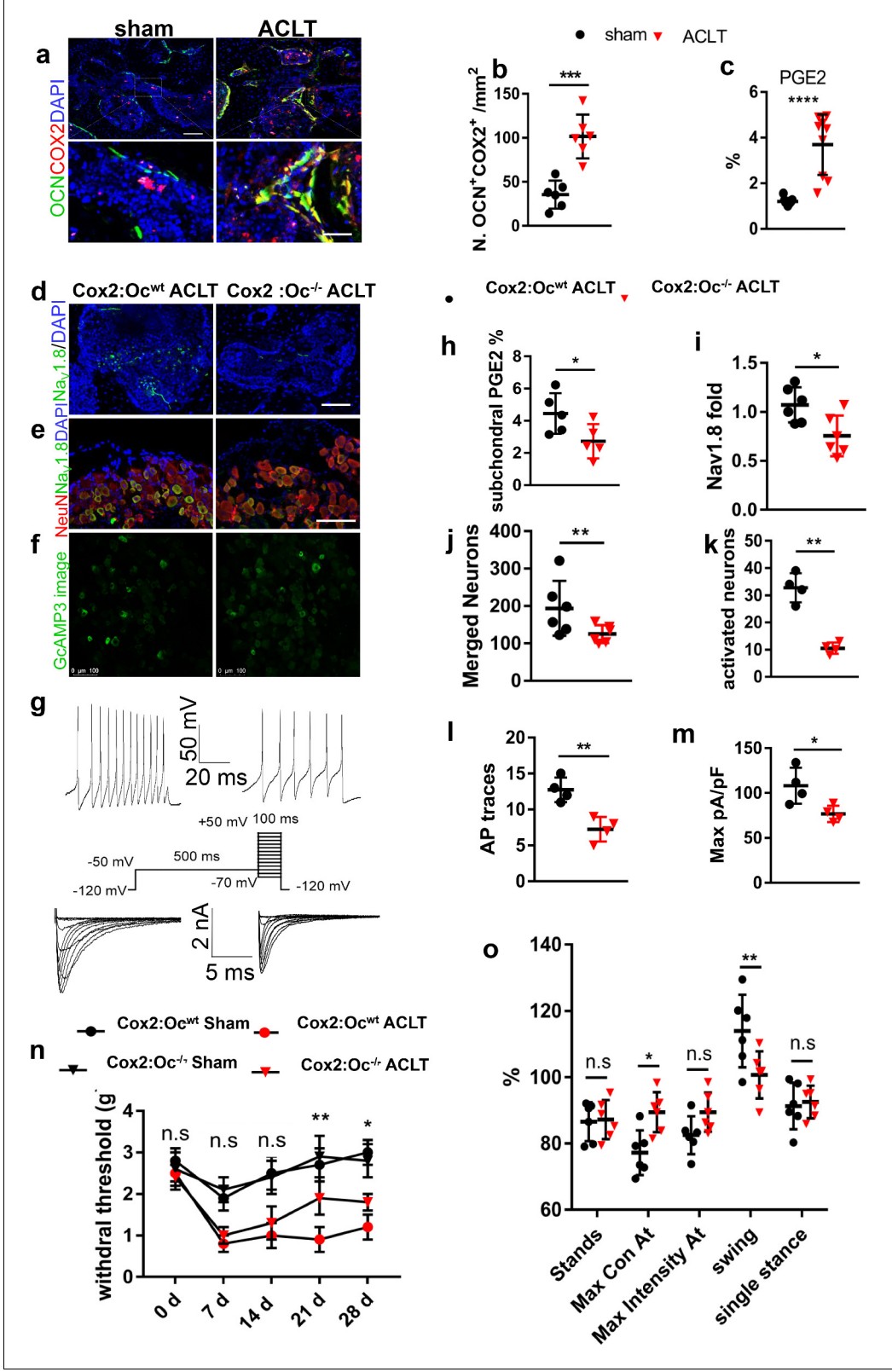

**Figure 2.** Decreased Na$_V$1.8 expression and ameliorated OA progression in Cox2:OCN cKO ACLT mice. (**a,b**) Representative pictures (**a**) and statistical analysis (**b**) of OCN and Cox2 co-stained cells of murine tibial subchondral bone after sham or ACLT surgery and 1 m. Scale bars, 50 μm (left) and 10 μm (right), n = 6 per group. (**c**) Relative concentration of subchondral PGE2 compared with total protein concentration before and after ACLT. (**e–g**) Na$_V$1.8 (green) immunostaining in subchondral bone (**d**), NeuN (red), Na$_V$1.8 (green) and DAPI (blue) co-immunostaining in ipsilateral L4 DRG (**e**),

*Figure 2 continued on next page*

*Figure 2 continued*

Activated neurons in ipsilateral L4 DRG using in vivo Pirt-GCaMP3 imaging (**f**) and AP traces and Na$_V$ currents (**g**) after sham or ACLT surgery at 1 m. Scale bars, 20 μm (**h**), 100 μm (**e**, **f**). (**h–o**) Statistical analysis of subchondral PGE2 concentration (**h**), Na$_V$1.8 immunofluorescence signal in subchondral bone (**i**), number of NeuN, Nav1.8 co-immunostained neurons in ipsilateral L4 DRG (**j**), number of activated neurons in ipsilateral L4 DRG using in vivo Pirt-GCaMP3 imaging (**k**), AP traces (**l**) and Na$_V$ currents (**m**), Catwalk gait analysis (**n**) and left HPWT (**o**) after sham or ACLT surgery. n = 6 per group, *p<0.05, **p<0.01, ***p<0.001, ****p<0.0001 compared with the sham-operated group at different time points. Statistical significance was determined by multifactorial ANOVA WITH BONFERRONI POST HOC TEST (**h**) or unpaired Student's *t* test (**b**, **c**, **h–m and o**), and all data are shown as scattered plots with means ± standard deviations.

The online version of this article includes the following source data and figure supplement(s) for figure 2:

**Source data 1.** Raw data of OCN Cox2 costaining, subchondral PGE2, *GcAMP3* imaging, NeuN Nav1.8 merged neurons, electrophysiological recordings, Von Frey tests and catwalk analysis.

**Figure supplement 1.** Subchondral bone remodeling in Cox2:OCN cKO and EP4:Avil cKO mice after ACLT.

**Figure supplement 1—source data 1.** Full scan of western blots in *Figure 2—figure supplement 1i*.

## Excessive PGE2 secreted by osteoblasts modifies Na$_V$1.8 for OA

We then examined the mechanism of upregulation of Na$_V$1.8 expression during OA progression. To examine whether excessive PGE2 contributes to the upregulation of Na$_V$1.8, we firstly performed immunostaining of cyclooxygenase 2 (Cox2) in subchondral bone sections. Cox2 expression was significantly increased in subchondral bone and primarily in osteocalcin positive osteoblastic cells post ACLT mice compared with sham-operated mice (*Figure 2a and b*). Consistently, PGE2 concentration in subchondral bone increased about three times in OA mice relative to sham-operated mice (*Figure 2c*, *Figure 2—figure supplement 1a–d*). To examine if elevated PGE2 upregulates the expression of Na$_V$1.8 for the mechanical allodynia in OA, we generated osteoblast specific *Cox2* deficient mice (*Cox2$^{-/-}_{Oc}$* mice) by crossbreeding *Cox2$^{fl/fl}$* mice with *Bglap-Cre* mice. PGE2 concentration in subchondral bone was significantly lower in *Bglap-Cre::Cox2$^{fl/fl}$* mice compared with *Cox2$^{fl/fl}$* mice post ACLT (*Figure 2h*). Notably, the Na$_V$1.8 immunofluorescence intensity in subchondral bone immunostaining was also significantly reduced in *Bglap-Cre::Cox2$^{fl/fl}$* mice compared with *Cox2$^{fl/fl}$* mice (*Figure 2d and i*). In addition, the number of Na$_V$1.8$^+$ neurons in DRG was also significantly decreased (*Figure 2e and j*) in the *Bglap-Cre::Cox2$^{fl/fl}$* mice relative to *Cox2$^{fl/fl}$* mice. We then investigated whether a decrease of PGE2 alleviates OA pain. We crossed *Bglap-Cre::Cox2$^{fl/fl}$* mice with *Pirt$^{GCaMP3fl/-}$* mice to measure the DRG neuronal excitability in *Bglap-Cre::Cox2$^{fl/fl}$::Pirt$^{GCaMP3fl/-}$* mice post ACLT. Pirt-GCaMP3 DRG imaging showed that the number of excited neurons was significantly reduced in *Bglap-Cre::Cox2$^{fl/fl}$::Pirt$^{GCaMP3fl/-}$* mice compared with *Cox2$^{fl/fl}$::Pirt$^{GCaMP3fl/-}$* mice post ACLT (*Figure 2f and k*). Moreover, the single neuron excitability in ipsilateral L4 DRG was functionally tested by whole cell patch clamp electrophysiology. The whole cell current clamp revealed that the action potential firing number was significantly decreased in *Bglap-Cre::Cox2$^{fl/fl}$* mice compared with *Cox2$^{fl/fl}$* mice after ACLT (*Figure 2g and l*). Concurrently, the Na$_V$1.8 current density was reduced for about 40% recorded by the whole-cell voltage-clamp (*Figure 2g and m*). The mechanical allodynia was simultaneously attenuated in *Bglap-Cre::Cox2$^{fl/fl}$* mice relative to *Cox2$^{fl/fl}$* mice as revealed by the Von Frey behavior test (*Figure 2n*). Catwalk analysis also showed that the Maximal Contact At and Maximal Intensity of ipsilateral hind paw was significantly higher in *Bglap-Cre::Cox2$^{fl/fl}$* mice compared with *Cox2$^{fl/fl}$* mice post ACLT (*Figure 2o*). Thus, PGE2 derived from osteoblastic cells stimulates the pain hypersensitivity in OA mice likely by upregulating of Na$_V$1.8 in subchondral nociceptive neurons.

## PGE2 signals through the EP4 receptor to sensitize sensory nerves in OA subchondral bone

To examine whether EP4 at sensory neurons is the primary receptor that propagates PGE2 signals in upregulating Na$_V$1.8and OA pain, we specifically knocked out EP4, the skeletal pain related receptor for PGE2 (*Yoshida et al., 2002*), in peripheral sensory nerves by crossbreeding *Advillin-Cre (Avil-Cre)* (*Zurborg et al., 2011*) mice with *Ptger4$^{fl/fl}$* mice (*Ptger4* is the gene that encodes EP4 receptor). Consistently with *Bglap-Cre::Cox2$^{fl/fl}$* mice, the intensity of Na$_V$1.8 immunofluorescence was significantly reduced in *Avil-Cre::Ptger4$^{fl/fl}$* mice compared with *Ptger4$^{fl/fl}$* mice post-ACLT (*Figure 3a, d*, *Figure 2—figure supplement 1e–h*). We then further confirm this finding in DRG neurons that

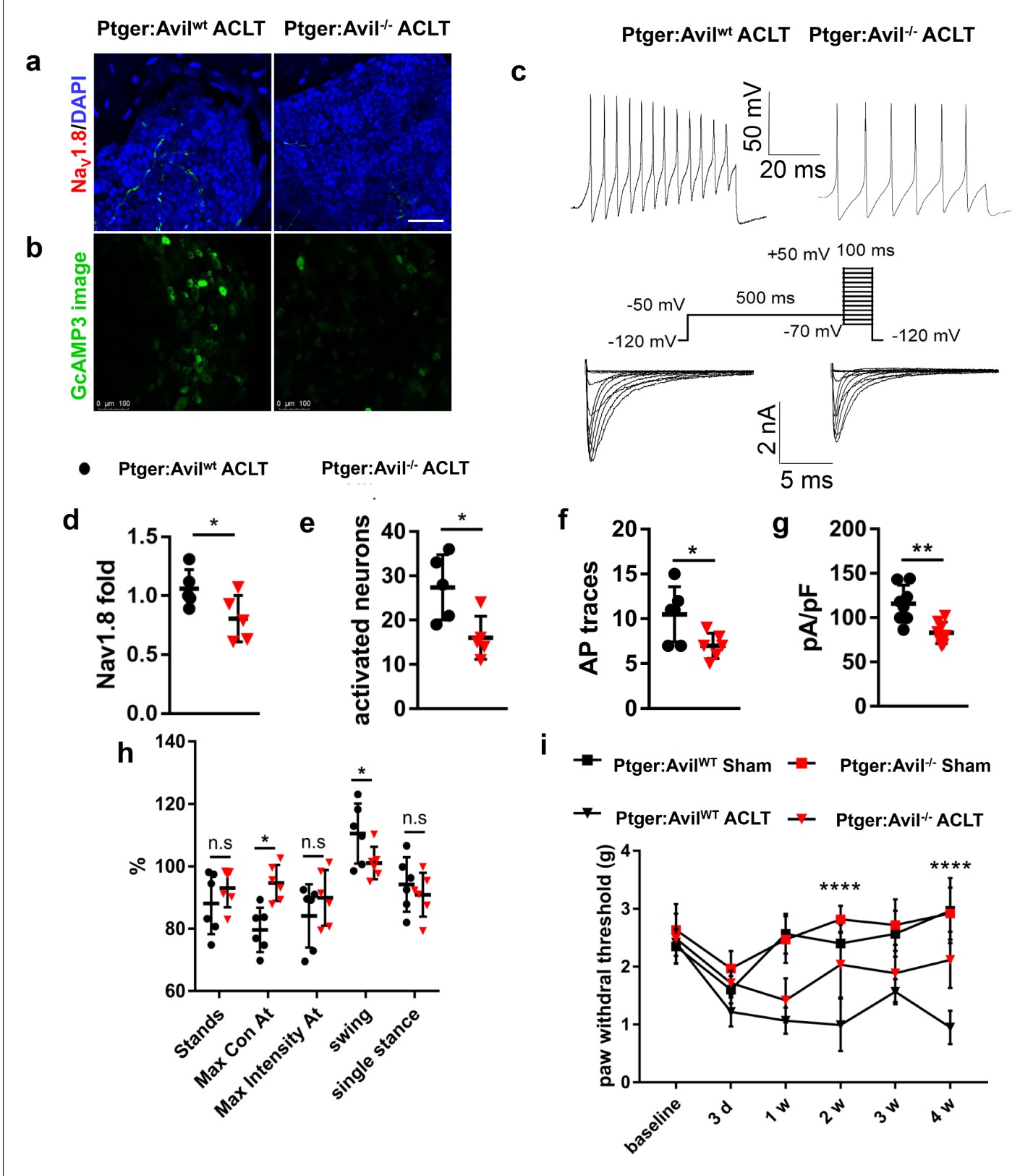

**Figure 3.** Decreased Na$_V$1.8 expression and ameliorated mechanical allodynia in *Avil-Cre::Ptger4$^{fl/fl}$* ACLT mice. (a, b) Na$_V$1.8 immunostaining in subchondral bone (a), Activated neurons in ipsilateral L4 DRG using in vivo Pirt-GCaMP3 imaging (b) after sham or ACLT surgery at 1 m. Scale bars, 20 μm (a), 100 μm (b). (c) Representative traces of action potentials (upper) and Nav1.8 currents (lower) of ipsilateral L3-5 DRG neurons after sham or ACLT

*Figure 3 continued on next page*

*Figure 3 continued*

surgery at 1 m. (**d–i**) Statistical analysis of Nav1.8 immunofluorescence signal in subchondral bone (**d**), number of activated neurons in ipsilateral L4 DRG using in vivo Pirt-GCaMP3 imaging (**e**), AP traces (**f**), max Nav1.8 current density (**g**), catwalk gait analysis (**h**) and left hindpaw PWT (**i**) after sham or ACLT surgery. n = 6 per group, *p<0.05, **p<0.01, ***p<0.001, ****p<0.0001 compared with the sham-operated group at different time points. Statistical significance was determined by multifactorial ANOVA WITH BONFERRONI POST HOC TEST (**i**) or unpaired Student's *t* test (**d–h**), and all data are shown as scattered plots with means ± standard deviations.

The online version of this article includes the following source data for figure 3:

**Source data 1.** Raw data of subchondral Nav1.8 fiber density, Von Frey tests, catwalk analysis, *GcAMP3* imaging, and electrophysiological recordings.

cultured in PGE2. We found that $Na_V1.8$ protein expression was significantly reduced by siRNA against EP4 in western blot analysis of levels from the cell lysates. Knocking-down the expression of EP1-EP3 did not have a significant effect on Nav1.8 expression (*Figure 2—figure supplement 1i*). To determine the effect of conditional deletion of EP4 on DRG neuronal excitability, we generated *Avil-Cre::Ptger4 $^{fl/fl}$::Pirt - GCaMP3$^{fl/-}$* and *Ptger4 $^{fl/fl}$::Pirt $^{GCaMP3fl/-}$* ACLT mice. In vivo ipsilateral L4 DRG *Pirt $^{GCaMP3}$* imaging demonstrated significantly dampened excitability in the *Ptger4 $^{fl/fl}$:: Pirt - GCaMP3$^{fl/-}$* mice relative to the control group (*Figure 3b and e*). The patch-clamp analysis further revealed that the DRG neuronal hypersensitivity and $Na_V1.8$ currents were significantly reduced in *Avil-Cre::Ptger4 $^{fl/fl}$* mice relative to *Ptger4 $^{fl/fl}$* mice post-ACLT (*Figure 3c,f,g*). Moreover, both Von Frey test and catwalk analysis showed attenuation of OA pain when EP4 is conditionally deleted in sensory neurons (*Figure 3h and i*). Thus, the EP4 receptor expressed in DRG neurons is responsible for the propagation of subchondral PGE2-induced upregulation of $Na_V1.8$ and neuronal excitability in OA mice.

## PGE2 stimulates $Na_V1.8$ transcription by inducing the binding of pCREB to $Na_V1.8$ promoter

To investigate the mechanism of PGE2 stimulated upregulation of $Na_V1.8$ expression, RT-qPCR was performed with mRNA isolated from primary DRG neurons treated with PGE2. The result showed that $Na_V1.8$ transcription levels were significantly elevated at 6 and 12 hr after incubation with PGE2 (*Figure 4a*). Moreover, PGE2 stimulated phosphorylation of protein kinase A (PKA) (*Gold et al., 1998*) and cAMP response element-binding protein (Creb1) (*Lonze and Ginty, 2002*; *Figure 4b*). Notably, the effect of PGE2 and Forskolin, a cAMP stimulant in the upregulation of $Na_V1.8$ protein expression was dampened by Creb1 inhibitor 666–15 (*Figure 4c*), indicating PGE2 stimulates Nav1.8 expression through the PKA-Creb1 signaling pathway. Consistently, the neuronal excitability and $Na_V1.8$ current density stimulated by PGE2 was abolished by the application of PKA inhibitor (PKI) or CREB inhibitor 666–15 (*Figure 4d–f*). Co-immunofluorescence staining further demonstrated that PKA levels significantly increased in $Na_V1.8$ positive DRG neurons in ACLT mice compared with sham-operated mice (*Figure 4g and h*). To examine the mechanism of PGE2-induced $Na_V1.8$ transcription, we performed chromatin immunoprecipitation (ChIP) assay with three potential pCreb1-binding elements in the $Na_V1.8$ promoter. ChIP assay revealed that pCreb1 binds to the $Na_V1.8$ promoter at binding site two to stimulate transcription of $Na_V1.8$ gene (*Figure 4i–k*). Taken together, our findings reveal that PGE2 induces transcription of $Na_V1.8$ by stimulation phosphorylation PKA and pCreb1, which directly binds to $Na_V1.8$ promoter.

## The deletion of $Na_V1.8$ in sensory nerve attenuates OA

We next tested whether the deletion of $Na_V1.8^+$ neurons could attenuate OA pain. *Scn10a* -**Cre** mice were crossed with *Rosa26$^{iDTRfl/fl}$* mice to generate *Scn10a -Cre:: Rosa26$^{iDTRfl/fl}$* mice. In these mice, the $Na_V1.8^+$ neurons underwent apoptosis upon receiving the injection of diphtheria toxin (*Buch et al., 2005*). The ablation of $Na_V1.8^+$ neurons had no effect on the articular cartilage deterioration, as shown by similar OARSI scores between *Scn10a -Cre:: Rosa26$^{iDTRfl/fl}$* mice and *Rosa26$^{iDTRfl/fl}$* mice post-ACLT (*Figure 5a,e*). The efficacy of specific neuron ablation was evidenced by a significant reduction of $Na_V1.8^+$ signals at both subchondral bone and DRG level (*Figure 5b,c,f,g*). Consistently, the DRG hypersensitivity was reduced as indicated by a decreased AP firing (*Figure 5d and h*). We further investigated whether the ablation of $Na_V1.8^+$ sensory neurons reduces the pain in OA mice by catwalk gait analysis (*Lakes and Allen, 2016*). The results showed that max intensity, which reflected mechanical pain sensitivity (*Kameda et al., 2017*), was increased in *Scn10a -Cre::*

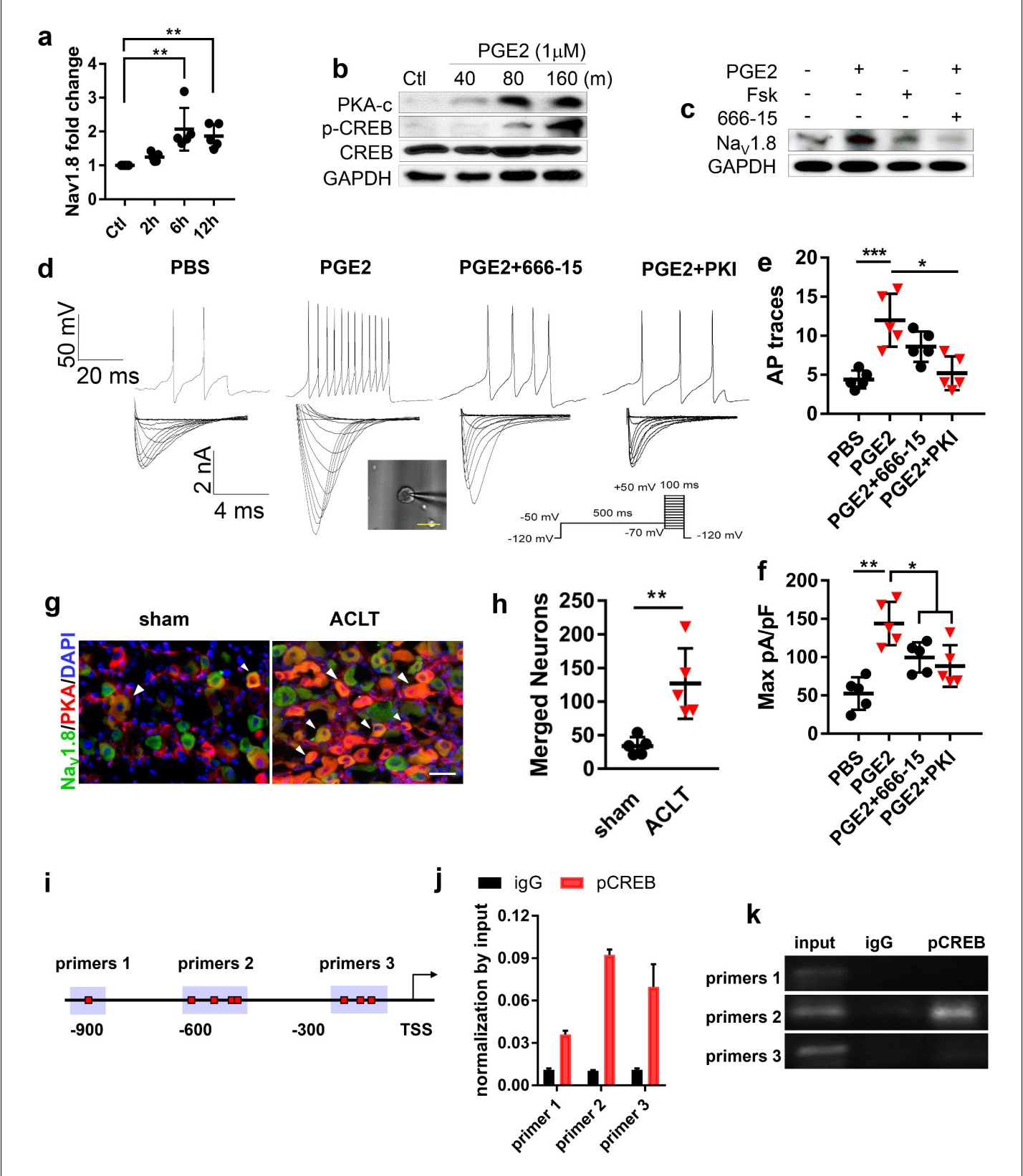

**Figure 4.** PGE2 upregulates Na$_V$1.8 through PKA signaling. (**a**) RT-QPCR analysis of Nav1.8 in cultured lumbar DRG neurons treated with PGE2 (1 μM) for 2–12 hr. n = 6 per group. (**b**) Representative of western blots of PKA-c, CREB and p-CREB in cultured lumbar DRG neurons treated with PGE2 (1 μM)

*Figure 4 continued on next page*

Figure 4 continued

for 40–160 min. (c) Representative of western blots of the Nav1.8 in cultured lumbar DRG neurons treated with PGE2 (1 µM), forskolin (10 µM), or 666–15 (1 µM) for 6 hr. Representative traces of action potentials (d, upper) and Nav1.8 currents (d, lower) and statistical analysis of maximal Nav1.8 current density (f) of cultured lumbar DRG neurons after sham or ACLT surgery at 1 m. n = 6 per group. (e, f) Co-immunostaining of NeuN and Nav1.8 (g) and statistical analysis of merged cell numbers (h) in ipsilateral L4 DRGs after sham or ACLT surgery at 1 m. n = 6 per group. (i–k) ChIP experiment showing putative primers (i), PCR (j) and gel running results (k) of Na$_V$1.8 promoter, the experiments were repeated three times. n.s, non significant, *p<0.05, **p<0.01, ***p<0.001, ****p<0.0001 compared with the sham-operated group at different time points. Statistical significance was determined by multifactorial ANOVA WITH BONFERRONI POST HOC TEST (a, e and f), unpaired Student's *t* test (h), all data are shown as scattered plots with means ± standard deviations.

The online version of this article includes the following source data for figure 4:

**Source data 1.** Raw data of Navs QPCR, subchondral Nav1.8 fiber density, Retrograde tracing, Von Frey tests, GcAMP3 imaging, and electrophysiological recordings.
**Source data 2.** Full scan of western blots in *Figure 4b*.
**Source data 3.** Full scan of western blots in *Figure 4c*.
**Source data 4.** Full scan of western blots in *Figure 4k*.

---

*Rosa26$^{iDTRfl/fl}$* mice (*Figure 5i*). Similarly, the Von Frey test displayed a significant reduction of HPWT in *Scn10a* -Cre:: *Rosa26$^{iDTRfl/fl}$* mice compared with *Rosa26$^{iDTRfl/fl}$* mice after ACLT (*Figure 5j*). Taken together, the *Scn10a* -Cre:: *Rosa26$^{iDTRfl/fl}$* mice indicates that specific ablation Na$_V$1.8 can alleviate OA pain in OA mice.

## Improvement of subchondral bone structure downregulates Na$_V$1.8 and attenuates OA progression

We previously showed that inhibition of excessive TGF-β activity attenuated OA progression by restring the coupling of subchondral bone remodeling (*Figure 6—figure supplement 1a–n*, *Figure 6—figure supplement 2* 4a-e). We have developed a small molecule drug by conjugating TGF-β type I receptor kinase inhibitor (TβR1I) covalently with alendronate through a metabolically cleavable carbamate linkage (*Qin et al., 2018*). The conjugate is effectively delivered to the bone surface where TβR1I is released by cleavage of the carbamate linkage in vivo. (*Figure 6—figure supplement 3a*). Administration of the conjugate in human MSCs effectively inhibited TGF-β signaling evidenced by a significant reduction of pSMAD2/3 (*Figure 6—figure supplement 3b and c*). As excessive PGE2 production and subsequent upregulation of Nav1.8 are triggered by abnormal bone remodeling, we investigated whether conjugate treatment can alleviate OA pain by downregulates the activity of Nav1.8. As expected, the articular cartilage degeneration was attenuated with a weekly injection of conjugate 100 ug/kg in ACLT mice compared with the vehicle group, with a significant improvement of the OARSI score (*Figure 6a and g*). In concurrent with the cartilage protection, the subchondral bone microarchitecture was improved in µCT analysis of ACLT mice treated with the conjugate treatment compared with the vehicle group (*Figure 6b and h*, *Figure 6—figure supplement 3f*). As shown in *Figure 6c and i*, phosphorylation of Smad2/3 was effectively inhibited by the conjugate in the subchondral bone. The number of TRAP$^+$ osteoclastic cells and Osterix$^+$ pre-osteoblast were reduced (*Figure 6—figure supplement 3e and g*, *Figure 6d and j*). As a result, the BML in tibial subchondral bone was significantly reduced in the conjugate treated group (*Figure 6f*) further indicating that coupling of the osteoclast bone resorption and osteoblastic bone formation were restored.

Finally, we examined whether the conjugate effect on the improvement of subchondral bone structure and articular cartilage degeneration could also relieve OA pain. Interestingly, subchondral PGE2 concentration was significantly reduced in ACLT mice with the conjugate treatment relative to the vehicle group (*Figure 6l*). Importantly, the expression of Na$_V$1.8 was also reduced in both subchondral bone (*Figure 6e and k*), and ipsilateral lumbar DRG (*Figure 6m*). Moreover, the electrophysiological tests demonstrate that data showed that conjugate treatment blunted the upregulation of DRG neuron activity (*Figure 6n and o*) and Na$_V$1.8 currents (*Figure 6n and p*) in ACLT mice. The effect of the conjugate on joint pain related behaviors were examined in the Catwalk test. HPWT, maximal contact AT and swing phase were significantly ameliorated in ACLT mice with the administration of conjugate relative to the vehicle group (*Figure 6q and r*). These data

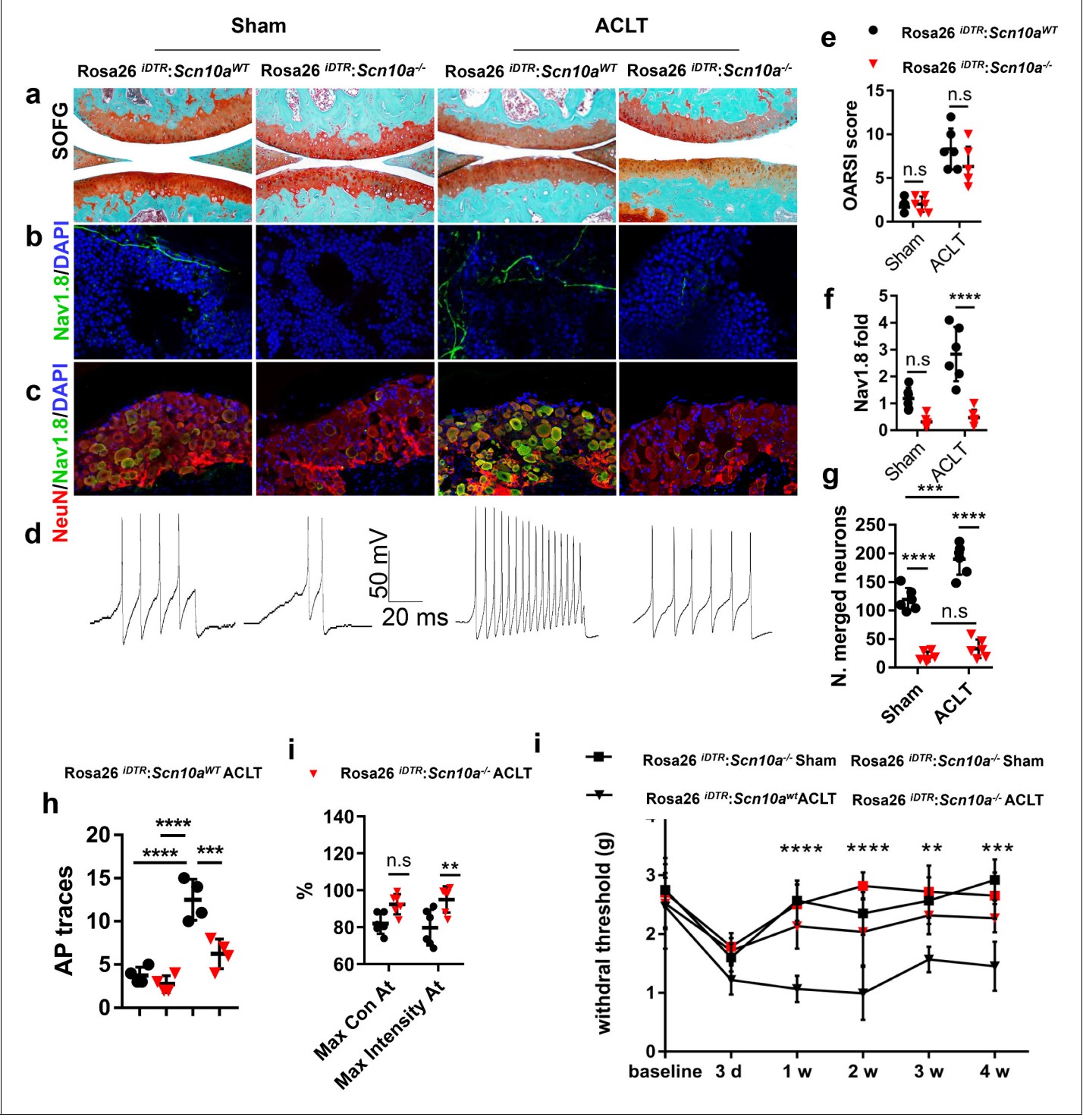

**Figure 5.** Mechanical allodynia is reduced in *Scn10a-Cre::Rosa26^iDTRfl/fl^* ACLT mice. (**a–d**) Representative photos of knee joint Safranin Orange and fast green staining (**a**), Na$_v$1.8 (green) and DAPI (blue) immunofluorescence in subchondral bone (**b**) and NeuN (red, Na$_v$1.8 (green) and DAPI (blue) co-immunostaining of ipsilateral L4 DRGs (**c**) and APs (**d**) after sham or ACLT surgery at 1 m. Scale bars, 500 µm (**a**), 20 µm (**b**) and 100 µm (**c**), n = 6 per group. (**e–j**) Statistical analysis of OARSI score (**e**), Nav1.8 immunofluorescence signal in subchondral bone (**f**), number of NeuN, Na$_v$1.8 co-immunostained neurons in ipsilateral L4 DRG (**g**), number of AP (**h**), catwalk gait analysis (**i**) and left hindpaw PWT (**j**) after sham or ACLT surgery. n = 6 per group, *p<0.05, **p<0.01, ***p<0.001, ****p<0.0001 compared with the sham-operated group at different time points. Statistical significance was determined by multifactorial ANOVA WITH BONFERRONI POST HOC TEST (**e–h and j**) or unpaired Student's *t* test (**i**), and all data are shown as scattered plots with means ± standard deviations.

*Figure 5 continued on next page*

*Figure 5 continued*

The online version of this article includes the following source data for figure 5:

**Source data 1.** Raw data of OARSI, subchondral Nav1.8 fiber density, NeuN nav1.8 costaining, Von Frey tests, catwalk analysis, GcAMP3 imaging, and electrophysiological recordings.

suggest that alendronate-TβR1I conjugate relieves OA pain by modifying the disease. This was likely achieved by the decrease of PGE2 in the improvement of subchondral bone structure.

## Discussion

Pain is the major symptom of OA, the most prevalent skeletal degenerative disease with no effective disease-modifying drugs. To date, the major local source and pathophysiological mechanisms of OA pain remain poorly understood, impeding the development of mechanism based strategies for OA pain attenuation. Based on clinical observations, in this study, we hypothesized that aberrant subchondral bone remodeling could be highly responsible for OA pain. Aberrant bone remodeling significantly stimulates the PGE2 production in subchondral bone, with the osteoblastic cell being the major source of production. Accordingly, OA pain alleviation can be achieved by specifically knocking out the PGE2 producing enzyme Cox2 in osteoblast or its receptor EP4 in peripheral sensory nerve, likely through reducing the expression of $Na_V1.8$. In particular, the direct ablation of $Na_V1.8^+$ DRG neurons demonstrates that $Na_V1.8$ overexpression at least partially mediates neuronal hypersensitivity in OA progression. Importantly, we demonstrated that pharmacologically inhibition of aberrant bone remodeling has superior treatment effect than purely blocking pain transduction pathway as evidenced by that conjugate improved subchondral bone structure, attenuated cartilage degeneration and ameliorated OA pain simultaneously while deleting $Na_V1.8^+$ neurons only alleviated OA pain without disease-modifying effect in OA pathologies.

Generally, central (*Kuner, 2010*) and peripheral sensitization (*Miller et al., 2017*) are two principal components for OA pain. Since surgical removal of a part of arthritic knee joint in total knee replacement can immediately relieve OA pain (*Skou et al., 2015*), it is believed that peripheral input is indispensable in OA pain sensitization. Several joint structures are plausible sources of OA pain (e. g., the synovium, tendons), but clinical tests do not reliably attribute the pain to those structures. Synovium, because of its dense innervation of sensory nerves, is thought to be one of the important sources of OA pain (*Kc et al., 2016*). Low grade of synovitis in OA could also be able to stimulate sensory nerve endings. However, human studies showed that synovial sensory nerve declined in some of the degenerative OA patients (*Dominique Muschter et al., 2017*; *Sellam and Berenbaum, 2010*), making this hypothesis inconclusive. Similarly, the increase of vascular and nerve growth in meniscus (*Ashraf et al., 2011*) and fat pad (*Bohnsack et al., 2005*) suggests that they might also be a source of pain. Several lines of clinical evidence point to the potential role of subchondral bone in the mechanical allodynia during OA progression (*Kwoh, 2013*; *Zhu et al., 2019*). This is clinically supported by the immediate pain relief in OA patient after removal of degraded cartilage and underlying subchondral bone in joint surgery (*Mittag et al., 2016*). Since articular cartilage is not innervated by sensory nerve, therefore, the densely innervated subchondral bone could be an essential local source for clinical OA pain. Identifying the main source of pain and related mechanisms is essential for the treatment of OA pain. The subchondral bone transmits mechanical loads produced by body weight and muscle activity. It is highly adaptable, with the ability to model and remodel in response to loading stresses. During OA development, the subchondral bone undergoes aberrant remodeling, leading to pathologic lesions. MRI studies have shown lower bone mineral density, also known as bone marrow lesions (BML), and more severe disruption of subchondral bone architecture in patients with OA (*Dore et al., 2009*; *Majumdar et al., 2004*). Subchondral BML is the first sign of OA in animal models (*Libicher et al., 2005*) and strongly correlate with knee pain in humans (*Davies-Tuck et al., 2009*). We previously have demonstrated aberrant bone remodeling of subchondral bone at the onset and pathological development of OA (*Zhen et al., 2013*). Studies have showed perivascular sensory and sympathetic nerve fibers breach the subchondral bone in OA compared to normal joint (*Mapp and Walsh, 2012*; *Walsh et al., 2010*). Recently, we found that

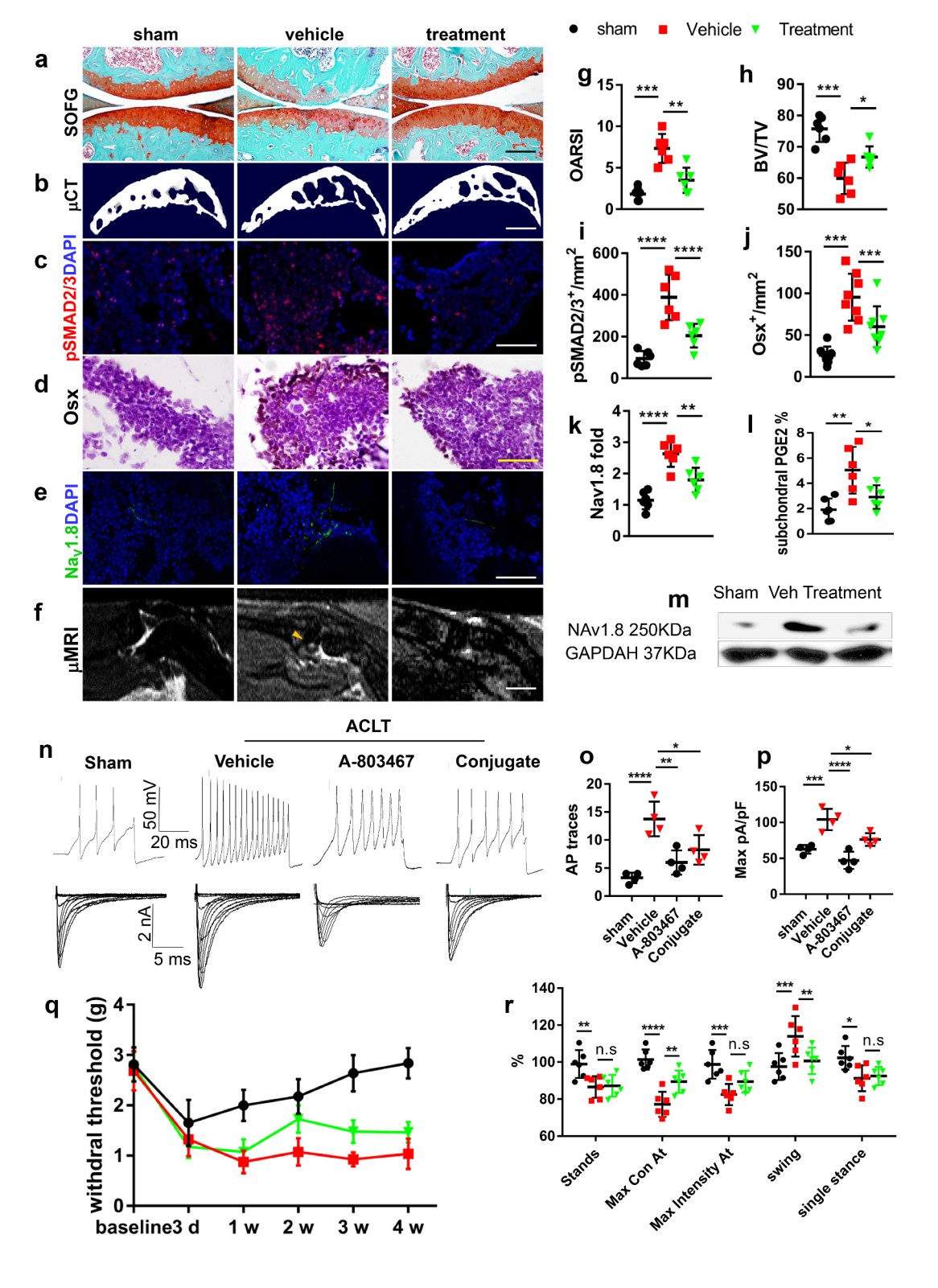

**Figure 6.** Targeting aberrant subchondral bone remodeling reduces Na$_V$1.8$^+$ innervation and ameliorates OA pain. (**a–f**) Representative photos of Safranin Orange and fast green staining (**a**), μCT 3D reconstruction (**b**), pSMAD2/3 (red) and DAPI (blue) immunostaining (**c**) Osterix immunostaining (**d**) Nav1.8 (green) and DAPI (blue) immunostaining (**e**) and T2 weighted fat suppression μMRI image showing bone marrow lesion (yellow arrows) (**f**) of murine tibial subchondral bone after sham or ACLT surgery at 1 m. Scale bars, 500 μm (**a**), 2 mm (**b**), 10 μm (**c, d**), 20 μm (**e**), and 5 mm (**f**), n = 6 per

*Figure 6 continued*

group. (g–l) Quantitative analysis of OARSI score (g), BV/TV (h), number of pSMAD2/3$^+$ cells per mm$^2$ (i) and number of Osterix$^+$ cells per mm$^2$ (j), relative pixel of Nav1.8 immunofluorescence signal (k) and subchondral PGE2 concentrations (l), after sham or ACLT surgery at 1 m. (m) Representative western blots of Na$_V$1.8 and GAPDH of ipsilateral L3-5 DRG lysate, experiments were repeated three times. (n–p) Representative traces of Aps (n upper), maximal current density (n lower), statistical analysis of AP numbers (o) and Na$_V$1.8 currents (p) of DRG 1 month post sham or ACLT. (q, r), left HPWT (q) and catwalk gait analysis (r) n = 6 per group, *p<0.05, **p<0.01, ***p<0.001, ****p<0.0001 compared with the sham-operated group at different time points. Statistical significance was determined by multifactorial ANOVA WITH BONFERRONI POST HOC TEST (j, k, L, m, n, o, q and r) or unpaired Student's *t* test (c), and all data are shown as scattered plots with means ± standard deviations.

The online version of this article includes the following source data and figure supplement(s) for figure 6:

**Source data 1.** Raw data of OARSI, microCT data, Osx, TRAP, pSMAD2/3, subchondral Nav1.8 fiber density, Retrograde tracing, Von Frey tests, GcAMP3 imaging, and electrophysiological recordings.
**Source data 2.** Full scan of western blots in *Figure 6m*.
**Figure supplement 1.** Aberrant subchondral bone remodeling after ACLT.
**Figure supplement 2.** Parameters of aberrant subchondral bone remodeling in human and mice osteoarthritis.
**Figure supplement 3.** Alendronate-TβR1I inhibitor conjugate attenuates aberrant subchondral bone remodeling after ACLT.

excessive Netrin-1 secreted by osteoclasts in subchondral bone induces sensory nerve axonal growth in OA (*Zhu et al., 2019*). We also found that during bone remodeling, PGE2, produced from arachidonic acid by the enzymatic activity of Cox2, activates EP4 in sensory nerves. In the present study, we found that the abnormal bone remodeling and temporary decrease of bone density in subchondral bone at the early stage of OA resembles the pathological changes as seen in osteoporosis. This explains why Cox2 activity and subsequent PGE2 production increased in response to the structural changes in OA subchondral bone. The increased nociceptive innervation to OA subchondral bone secondary to excessive netrin-1 secretion by the osteoclasts therefore favors the PGE2 induced neuronal excitations. Taken together, we believe that the development of OA pain is a synergistic result involving central sensitization in spinal cord in conjunction with peripheral input from subchondral bone, synovium, meniscus and fat pad etc.

The molecular mechanism of neuronal sensitization remains a poorly understood facet of OA pathophysiology. It is widely accepted that neuronal plasticity including activation, transcriptional modification and post-transcriptional of ion channels related to electrical excitability could contribute to the generation of chronic pain (*Woolf and Salter, 2000*). Giving the essential role of action potential firing in peripheral nerve input, the possible involvement of ion channels was investigated in recent studies. Our findings suggest that Na$_V$1.8 is the most upregulated Na$_V$ channel with a restricted localization in DRG. We therefore focused on Na$_V$1.8 to the possible molecular events based on the extensive over-activation of subchondral bone remodeling in OA progression. We found the expression rate of Na$_V$1.8 was significantly elevated in subchondral bone marrow, sciatic nerve and ipsilateral lumbar DRG levels. And the further screening analysis showed the expression rate of Na$_V$1.8 mainly elevated in CGRP$^+$ nociceptive fiber and piezo2$^+$ low threshold mechanoceptive fibers. This pattern of modification of Na$_V$1.8 expression in sensory neurons could explain the high sensitivity in polymodal nociception and mechanoception after ACLT. Functionally, this upregulation of expression led to larger Na$_V$1.8 currents and higher excitability of DRG neurons after ACLT. The Na$_V$1.8 currents are thought to be essential for action potential firing at the initial state. Being activated by PGE2, the lager Na$_V$1.8 currents could make the DRGs easier for action potential firing, thus transmitting the pain signals into higher centers for mechanical allodynia. In the short term, phosphorylation of Na$_V$1.8 by PGE2 may increase the inward currents by opening the Na$_V$1.8 ion gating mechanism (*Hudmon et al., 2008*). Also, PGE2 increases the expression of Na$_V$1.8 in a relatively long term of stimulation by PKA signaling. However, the role of other modalities of modulations like phosphorylation, methylglyoxalation need to be investigated in future studies. Nevertheless, future studies should be conducted to explain how this elevation of Na$_V$1.8 could be integrated and translated into the central nervous system as pain signals.

Pain sensation and OA progression are often dissociated. Late stage of radiographic OA patients with extensive subchondral bone sclerosis may result in less pain sensation and early stage patients with significant subchondral BML can be very painful. Moreover, the anti-nerve growth factor (NGF) tanezumab administration to OA patients relief pain with no significant attenuation on OA cartilage

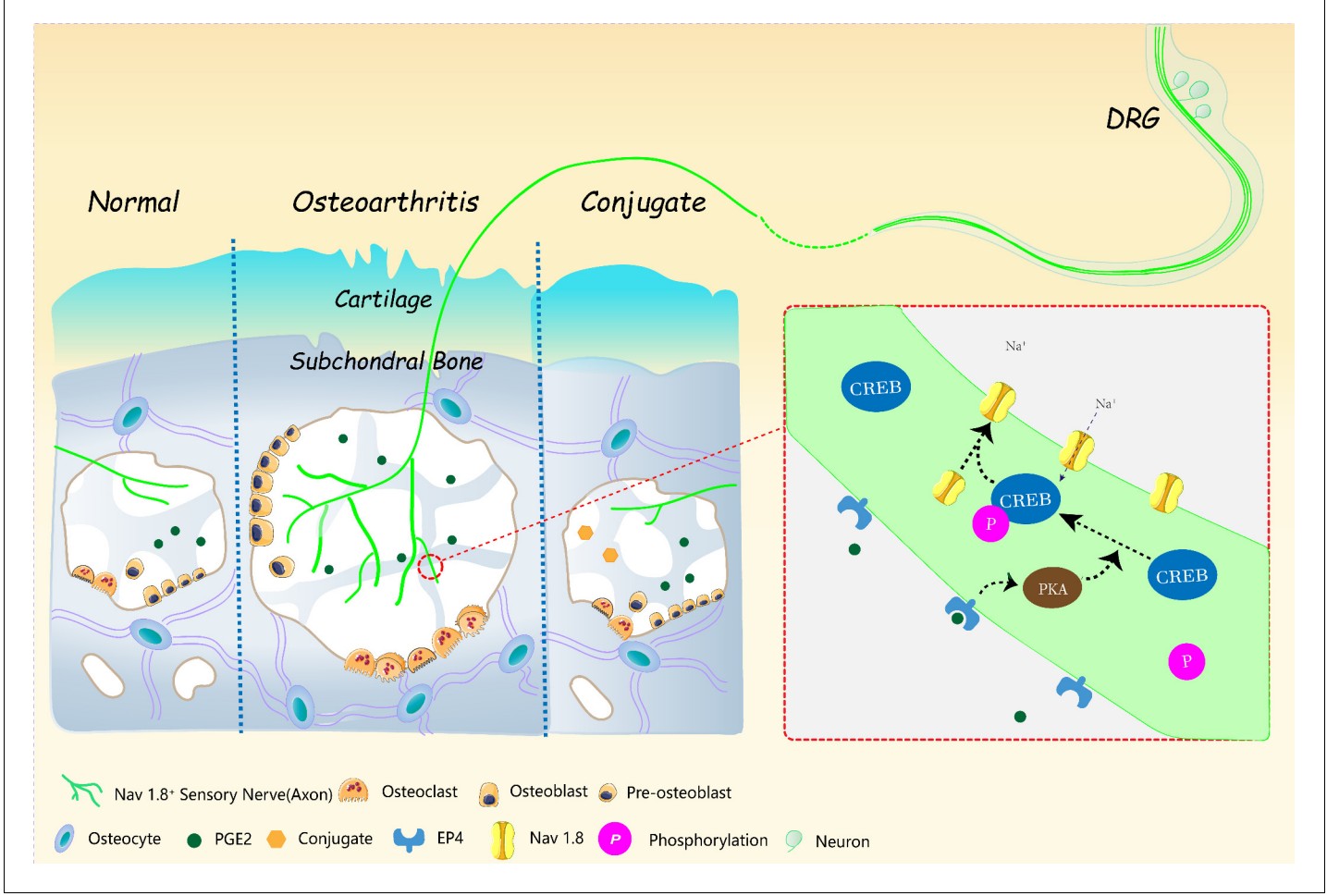

**Figure 7.** The working model of osteoblastic PGE2 induces OA progression by Na$_V$1.8 modification.

protection or subchondral bone sclerosis (*Lane and Corr, 2017*). Consistently, although pain relief can be achieved to some extent by targeting Na$_V$1.8, the cartilage or subchondral bone was not significantly protected in OA progression after ACLT in our study. These results indicate a comprehensive therapy for OA should target upstream events that cause OA progression and pain. Here, we provide a proof of principle for the potential of small molecule drug balancing aberrant bone remodeling to attenuate mechanical allodynia and OA progression in general. We achieved the bone-targeted TGF-β inhibition by conjugating the TGF-β type I receptor inhibitor (LY-2109761 with an osteoclast targeting drug alendronate. Consistent with our previous findings, the conjugate rebalanced the uncoupled subchondral bone remodeling and reduced over-activated osteoblastic bone formation through targeting aberrant TGF-β signaling in ACLT mice model. This rebalance was effective in pain alleviation through a reduction of excessive PGE2 released into subchondral bone marrow and down-regulation of Na$_V$1.8 expression and electric property to reduce mechanical allodynia in OA. Meanwhile, the reconstruction of subchondral bone architecture protected the overlying cartilage destruction and delayed the progression of the overall OA process (*Figure 7*). We believe this mechanism based management of OA pain may shed light on the strategy of various musculoskeletal disorders with chronic pain symptom. Nevertheless, because the pharmacological or toxicological aspect of the conjugate in vivo are largely unknown, further enhancement of therapeutic effect may be achieved with optimization of dosing and delivery strategies.

# Materials and methods

## Key resources table

| Reagent type (species) or resource | Designation | Source or reference | Identifiers | Additional information |
|---|---|---|---|---|
| Strain, Strain background (*Mus musculus*) | *Ptger4*floxed | (*Chen et al., 2019*) | N/A | C57BL/6 background |
| Strain, Strain background (*Mus musculus*) | *Bglap-Cre* | (*Tomlinson et al., 2016*) | N/A | C57BL/6 background |
| Strain, Strain background (*Mus musculus*) | Pirt-GcaMP3 floxed | (*Kim et al., 2008*) | N/A | C57BL/6 background |
| Strain, Strain background (*Mus musculus*) | *Advillin-Cre(Avil-Cre)* | (*Zurborg et al., 2011*) | N/A | C57BL/6 background |
| Strain, Strain background (*Mus musculus*) | *Rosa26*iDTRfloxed | Jackson Laboratory | C57BL/6-Gt (ROSA)26Sortm1 (HBEGF)Awai/J Stock No: 007900 | C57BL/6 background |
| Strain, Stran background Rattus norvegicus | Sprague Dawley (SD) | Charles River | N/A | |
| Strain, Strain backgrund (*Mus musculus*) | *Scn10a-Cre* | (*Duan et al., 2018*) | N/A | C57BL/6 background |
| Strain, Strain backgrund (*Mus musculus*) | *Cox2* floxed | Harvey Herschman | N/A | C57BL/6 background |
| Sequence-based reagent | *Scn10a-Cre* forward | | PCR Primer | 5′-TGTAGATGGACTGCAGAGGATGGA-3′ |
| Sequence-based reagent | *Scn10a-Cre* reverse | | PCR Primer | 5′-AAATGTTGCTGGATAGTTTTTACTGCC-3′ |
| Sequence-based reagent | Pirt-GCaMP3 fl primer 1 | | PCR Primer | 5′-TCCCCTCTACTGAGAGCCAG-3′ |
| Sequence-based reagent | Pirt-GCaMP3fl primer 2 | | PCR Primer | 5′-GGCCCTATCATCCTGAGCAC-3′ |
| Sequence-based reagent | Pirt-GCaMP3fl primer 3 | | PCR Primer | 5′-ATAGCTCTGACTGCGTGACC-3′ |
| Sequence-based reagent | Avil-Cre: forward | | PCR Primer | 5′-CCCTGTTCACTGTGAGTAGG-3′ |
| Sequence-based reagent | Avil-Cre: reverse | | PCR Primer | 5′-GCGATCCCTGAACATGTCCATC-3′ |
| Sequence-based reagent | Avil-Cre: wildtype | | PCR Primer | 5′-AGTATCTGGTAGGTGCTTCCAG-3′ |
| Sequence-based reagent | Bglap-Cre: forward | | PCR Primer | 5′-CAAATAGCCCTGGCAGATTC-3′ |
| Sequence-based reagent | Bglap-Cre: reverse | | PCR Primer | Reverse: 5′-TGATACAAGGGACATCTTCC-3′ |
| Sequence-based reagent | Cox2 loxP allele forward: | | PCR Primer | 5′-AATTACTGCTGAAGCCCACC-3 |
| Sequence-based reagent | Cox2 loxP allele reverse | | PCR Primer | 5′-GAATCTCCTAGAACTGACTGG-3′ |
| Sequence-based reagent | *Ptger4* loxP allele forward | | PCR Primer | 5′-TCTGTGAAGCGAGTCCTTAGGCT-3′ |
| Sequence-based reagent | *Ptger4* loxP allele reverse | | PCR Primer | 5′-CGCACTCTCTCTCTCCCAAGGAA-3′ |

*Continued on next page*

*Continued*

| Reagent type (species) or resource | Designation | Source or reference | Identifiers | Additional information |
|---|---|---|---|---|
| Sequence-based reagent | *Rosa26*iDTRfloxed forward | | PCR Primer | 5'-GCGAAGAGTT TGTCCTCAACC-3' |
| Sequence-based reagent | *Rosa26*iDTRfloxed reverse | | PCR Primer | 5'-AAAGTCGCTCT GAGTTGTTAT-3' |
| Sequence-based reagent | Gapdh forward | | RT-PCR Primer | 5'-TCCATGACAAC TTTGGCATTG-3' |
| Sequence-based reagent | Gapdh reverse | | RT-PCR Primer | 5'-CAGTCTTCTGG GTGGCAGTGA-3' |
| Sequence-based reagent | Scn1a forward | | RT-PCR Primer | 5'-AACAAGCTTGAT TCACATACAATAAG-3' |
| Sequence-based reagent | Scn1a reverse | | RT-PCR Primer | 5'-AGGAGGGCGGA CAAGCTG-3' |
| Sequence-based reagent | Scn2a forward | | RT-PCR Primer | 5'-GGGAACGCCCA TCAAAGAAG-3' |
| Sequence-based reagent | Scn2a reverse | | RT-PCR Primer | 5'-ACGCTATCGTA GGAAGGTGG-3' |
| Sequence-based reagent | Scn3a forward | | RT-PCR Primer | 5'-AGGCATGAGGGT GGTTGTGAACG-3' |
| Sequence-based reagent | Scn3a reverse | | RT-PCR Primer | 5'-CAGAAGATGAG GCACACCAGTAGC-3' |
| Sequence-based reagent | Scn8a forward | | RT-PCR Primer | 5'-AGTAACCCTCCA GAATGGTCCAA-3' |
| Sequence-based reagent | Scn8a reverse | | RT-PCR Primer | 5'-GTCTAACCAGT TCCACGGGTCT-3' |
| Sequence-based reagent | Scn9a forward | | RT-PCR Primer | 5'-TCCTTTATTCATAA TCCCAGCCTCAC-3' |
| Sequence-based reagent | Scn9a reverse | | RT-PCR Primer | 5'-GATCGGTTCCG TCTCTCTTTGC-3' |
| Sequence-based reagent | Scn10a forward | | RT-PCR Primer | 5'-ACCGACAATCAG AGCGAGGAG-3' |
| Sequence-based reagent | Scn10a reverse | | RT-PCR Primer | 5'-ACAGACTAGAAA TGGACAGAATCACC-3' |
| Sequence-based reagent | Scn11a forward | | RT-PCR Primer | 5'-TGAGGCAACACTAC TTCACCAATG-3' |
| Sequence-based reagent | Scn11a reverse | | RT-PCR Primer | 5'-AGCCAGAAACCAA GGTACTAATGATG-3' |
| Sequence-based reagent | Creb1 forward | | ChIP-PCR Primer 1 | 5'-AGTATGGTCCTTCG TGGAATACCAG-3' |
| Sequence-based reagent | Creb1reverse | | ChIP-PCR Primer 1 | 5'-GCTATACTGCAG GAAACTGGCGA-3' |
| Sequence-based reagent | Creb1forward | | ChIP-PCR Primer 2 | 5'-AGCTCCCTTCTC AGCTCTCAC-3' |
| Sequence-based reagent | Creb1reverse | | ChIP-PCR Primer 2 | 5'-CAATCTACCCAGT CTCCCTCTTTGG-3' |
| Sequence-based reagent | Creb1forward | | ChIP-PCR Primer 3 | 5'-GAGCACCATCC AGCAAGCAG-3' |
| Sequence-based reagent | Creb1reverse | | ChIP-PCR Primer 3 | 5'-CCAGCTCTGCG AAACTTACACT-3' |
| Antibody | Rabbit polyclonal Anti-Nav1.8 | Alomone Labs | ASC-016, | 1:50 |
| Antibody | Rabbit polyclonal Anti-pSmad2/3 | Santa Cruz Bio | sc-11769 | 1:50, |

*Continued on next page*

*Continued*

| Reagent type (species) or resource | Designation | Source or reference | Identifiers | Additional information |
|---|---|---|---|---|
| Antibody | Rabbit polyclonal Anti-Osterix | Abcam | ab22552 | 1:300 |
| Antibody | Rabbit polyclonal Anti- Osteocalcin | Takara bio Inc, | M173 | 1:200 |
| Antibody | Rabbit polyclonal Anti- Cox2 | Abcam | ab15191 | 1:100 |
| Antibody | Mouse monocloncal anti-PKA-c | Abcam | ab75991 | 1:200 |
| Software | Graphpad 8.0 | | Statistical Analysis | graph preparation, statistical analysis |

## Animals

We purchased C57BL/6J (WT) 3 months old male mice from Jackson Laboratories. We purchased Sprague Dawley (SD) 3 months old male rats from Charles River company. To develop the mechanical instability related OA model, we used ACLT surgery (*Malfait and Little, 2015*). Briefly, after ketamine and xylazine anesthesia, the left ACL was surgically transected and sham operations were performed on other groups of mice. For the time-course experiments, mice were euthanized at 4, 8 or 12 weeks after surgery (n = 6 per group).

The *Rosa26$^{iDTRfl/fl}$* mice were purchased from Jackson Laboratory. The *Advillin-Cre* (*Avil-Cre*) and **Pirt$^{GCaMP3}$** mouse strain were kindly provided by Xingzhong Dong (The Johns Hopkins University). The *Bglap-Cre* mice were provided by Thomas J. Clemens (The Johns Hopkins University). The *Cox2$^{fl/fl}$* mice were kindly provided by Harvey Herschman (University of California, Los Angeles). The *Ptger4 $^{fl/fl}$* mice were provided by Brian L. Kelsall (the National Institutes of Health). The *Scn10a-Cre* mice were kindly provided by Yun Guan (The Johns Hopkins University). Heterozygous *Bglap-Cre* mice were crossed with a *Cox2$^{fl/fl}$* mouse; the offspring were intercrossed to generate the following genotypes: WT, *Bglap-Cre*, *Cox2$^{fl/fl}$*, *Bglap-Cre::Cox2$^{fl/fl}$*. *Cox2$^{fl/fl}$* or *Bglap-Cre::Cox2$^{fl/fl}$* **mice were further crossed with *Pirt$^{GCaMP3fl/-}$* mice to generate *Cox2$^{fl/fl}$::Pirt$^{GCaMP3fl/-}$* mice or *Bglap-Cre:: Cox2::Pirt$^{GCaMP3\ fl/-}$* mice for in vivo *GCaMP3* DRG imaging.** Heterozygous *Avil-Cre* mice were crossed with *Ptger4$^{fl/fl}$* mice. The offspring were intercrossed to generate the following genotypes: wild type (referred as 'WT' in the text), *Avil-Cre* (Cre recombinase expressed driven by Advillin promoter), *Ptger4$^{fl/fl}$*, *Avil-Cre:: Ptger4$^{fl/fl}$* (conditional deletion of EP4 receptor in Advillin lineage cells). *Ptger4$^{fl/fl}$* or *Avil-Cre:: Ptger4$^{fl/fl}$* mice were further crossed with *Pirt-GCaMP3 $^{fl/-}$* mice to generate *Ptger4$^{fl/fl}$::Pirt-GCaMP3$^{fl/-}$* mice or *Avil-Cre:: Ptger4$^{fl/fl}$::Pirt$^{GCaMP3\ fl/}$* mice for in vivo *GCaMP3* DRG imaging. Heterozygous *Scn10a-Cre* mice were crossed with the *Rosa26$^{iDTRfl/f}$* mouse; the offspring were intercrossed to generate the following genotypes: WT, *Scn10a-Cre::Rosa26$^{iDTRfl/fl}$*, *Scn10a-Cre::Rosa26$^{iDTRfl/fl}$* mice. We injected 12-week-old *Scn10a-Cre:: Rosa26$^{iDTRfl/fl}$* or *Rosa26$^{iDTRfl/fl}$* mice with 1 µg/kg DTX intraperitoneally three times per week after ACLT for 4 weeks. We obtained femurs, tibiae and DRG from the mice after euthanasia. For conjugate injections, we used intraperitoneal injection method and 1 mg/kg per week dosage according to previous toxicological experiments. All animals were maintained at the animal facility of The Johns Hopkins University School of Medicine. All the experimental protocols were approved by the Animal Care and Use Committee of The Johns Hopkins University (Protocol number: Mo18M308).

## Human samples

After approval by the Institutional Review Board of The Johns Hopkins Hospital,, we collected tibial plateau specimens from eight individuals with osteoarthritis that underwent total knee arthroplasty. The knee joints from three healthy young adults underwent lower limb amputations after trauma serves as healthy controls. The demographic data of patients were collected. The samples were used to perform histology and immunohistochemistry after decalcification.

## Histology

Immediately after euthanasia, we resected and fixed the animals knee joints or DRG in 10% buffered formalin for 24 hr, decalcified them in 0.5 M ethylenediaminetetraacetic acid (EDTA, pH 7.4) for 14 d and embedded them in paraffin or gelatin solution (20% D-sucrose, 2% Polyvinylpyrrolidone (PVP) and 8% gelatin in PBS). Four-micrometer sagittal oriented sections of the medial compartment of left knees were processed for hematoxylin and eosin, safranin orange and fast green and Tartrate-resistant acid phosphatase (TRAP) staining (Sigma). For immunohistology and immunofluorescence, slides (4 μm for immunohistology, 20 μm for DRG, 60 μm for knee immunofluorescence) were incubated with antigen retrieval buffer (Dako, S169984-2) at 96℃ for 15 min, gradually cooled to room temperature and washed with tris-buffered saline with Tween (TBST). After blocking, the slides were incubated with primary antibodies overnight at 4℃. Secondary antibody (1:200) was used to incubate the samples for 1 hr at room temperature. For immunohistochemical staining, a horseradish peroxidase–streptavidin detection system (Dako) was used to detect immunoactivity, followed by counterstaining with hematoxylin (Sigma-Aldrich). For immunofluorescence, the fluorescent conjugated secondary antibody (1:200) was applied. The photographs of the immunohistology sections were recorded by light microscopy (DP71 microscope camera, Olympus) and analyzed by OsteoMeasure XP software (OsteoMetrics). We calculated OARSI scores as previously described (Glasson et al., 2010). The OARSI scores were evaluated by two independent graders and the averages were taken. For the immunofluorescence, the photographs were shot under laser confocal microscopy (Zeiss, LSM 780) and Zen 2.2 software.

## μCT and in vivo μMRI

The mice knees were scanned using high-resolution μCT (SkyScan 1275, Bruker microCT) as previously described (Zhen et al., 2013). The scanner was set at a voltage of 65 kVp, a current of 153 μA and a resolution of 5.7 μm per pixel. We reconstructed and analyzed outcomes using NRecon v1.6, and CTAn v1.9, respectively. Three-dimensional reconstructions were done by CTVol v2.0 (Bruker microCT). We defined the region of interest to cover the trabecular part of the medial compartment of tibial subchondral bone, and five consecutive images from the medial tibial plateau were used for 3-dimensional reconstruction. We analyzed 3D parameters as following: TV (total tissue volume; containing both trabecular and cortical bone), BV/TV (trabecular bone volume per tissue volume) and Tb.Pf (trabecular pattern factor).

We performed in vivo μMRI studies on a horizontal 9.4T Bruker Biospec preclinical scanner according to our previous protocol (Zhen et al., 2013). Briefly, we showed subchondral BML by T2-weighted scanning with 2D RARE (rapid acquisition with relaxation enhancement) sequence, a TE/TR (echo time/repetition time) of 15.17 ms/3,000 ms, 30 slices at 0.35 mm thickness, 1.75 cm ×1.75 cm field of view (FOV) with a matrix size of 256 × 128. The fat suppression was done in T2-weighted imaging with a chemical shift selective fat saturation pulse tuned to the fat resonant frequency.

## Cell culture

Bilateral lumbar DRGs were harvested from 4 week male WT mice. For DRG neuron culture medium, MEM was supplemented with 5% fetal bovine serum (Gibco), 2X penicillin and streptomycin solution (Gibco), 1X GlutaMAX (Thermo Fisher), 20 μM 5-fluoro-2-deoxyuridine (Sigma-Aldrich) and 20 μM uridine (Sigma-Aldrich). DRG neurons were digested and dissociated with 1 mg/ml collagenase D (Roche) for 90 min and then 1X TrypLE Express solution (Thermo Fisher) for 15 min. The dissociated DRG neurons were placed on a precoated dish with 100 μg / ml poly-D-lysine (thermal fisher) and 10 μg / ml laminin (thermal fisher). 100 ng / ml Nerve growth factor (R and D) was applied to maintain the neuronal activity. After 24 hr incubation, PGE2 (1 μM) or PBS were applied to stimulate the DRG neurons. In vitro RNA interference was performed using commercially available RNAi products from Thermal Scientific (s72365, s72370, s72373, and s72375) and the protocol was followed by the manufacture's instruction. Briefly, after neuron seeding for 24 hr, media were replaced for the cells to be prepared for transfection. Lipofectamine RNAi MAX (13778100, Invitrogen) was diluted in OptiMEM (31985062, Thermal Fisher) and incubated for 5 min, then mixed with siRNAs or scramble control RNAs for five mins. Diluted DNA and Lipofectamine RNAi MAX were mixed and incubated at room temperature for 20 min and then used to transfect the DRG neurons. The medium was replaced 10 hr following transfection and neurons were harvested 24 hr after transfection. Similarly, the human

GFP labeled MSC was purchased from Cyagen and is cultured in MEM with 10% fetal bovine serum (Gibco), 1X penicillin and streptomycin solution (Gibco).

## In vivo Pirt-GCaMP3 DRG imaging

We used Pirt-GCaMP3$^{f/-}$ mice in DRG imaging. In order to monitor the activity of large populations of DRG neurons in intact live animals, Dr Xinzhong Dong's Lab developed an in vivo imaging technique by using $Pirt^{GCaMP3}$ genetically engineered mice, in which the genetic-encoded Ca$^{2+}$indicator $GCaMP3$ is specifically expressed in >95% of all DRG neurons by Pirt promoter (*Kim et al., 2008*; *Kim et al., 2014b*).After surgical exposure of ipsilateral L4 DRG, in vivo imaging was immediately performed. Similarly as previously described (*Miller et al., 2018*), the animals were maintained under inhalation anesthesia with assisted ventilation through endotracheal incubation. A laser scanning confocal microscope (Leica LSI microscope system) with a water immersed lens was used to capture the fluorescent signals. Live images were acquired at 10 frames with 600 Hz in frame-scan mode per 6–7 s, at depths below the dura ranging from 0 to 70 μm. 25 g of direct compression was applied to the ipsilateral knee after ACLT or sham surgery using a rodent pincher (IITC Life Science) to stimulate DRG neuronal firing. The duration of the mechanical force application maintained 15–30 s after 40–50 s of baseline imaging and the activated neuron number was counted and analyzed.

## Electrophysiology

Whole-cell current-clamp recordings were performed to perform the action potential of DR neurons according to the previous study (*Bierhaus et al., 2012*). Only small and medium-sized DRG neurons with a resting membrane potential more negative than −40 mV were recorded. The extracellular solution contained (in mM): NaCl 140, KCl 4, CaCl$_2$ 2, MgCl$_2$ 1; HEPES 10, NaOH 4.55, glucose 5 (pH 7.4, 300–310 mOsm/kg H$_2$O). The pipette solution contained (in mM): KCl 135, MgCl$_2$ 0.1, Mg-ATP 1.6, HEPES 10, EGTA 2, (pH 7.3 at 25℃, adjusted with NaOH). The voltage was firstly clamped at −60 mV. For action potential stimulation, the frequency is by 2 × and 3 × rheobase and ramp current stimulation (0.1, 0.3, 0.5, and 1.0 nA/sec ramp current).

To measure the TTX resistant Na$_V$1.8 currents in DRG neurons, the voltage-clamp technique was used. For recordings on DRG neurons, the extracellular solution contained (mM): NaCl 60, KCl 3, Choline-Cl 80, CaCl$_2$ 0.1, MgCl$_2$ 0.1, HEPES 10, tetraethylammonium chloride 10, glucose 10 and CdCl$_2$ 0.1 (pH adjusted to 7.4, 300–310 mOsm/kg H$_2$O) TTX (1 uM) and TC-N 1572 (1.6uM) were applied to the solution to block TTX sensitive sodium current and Na$_V$1.9 currents. The pipette solution contained (mM) CsF 140, EGTA 5, MgCl 1, and HEPES 10, glucose 10 (pH 7.4, osmolarity 285–295 mOsm/kg H$_2$O). Only cells with an initial seal >1 GΩ were recorded. The Na$_V$1.8 currents were recorded responding to potential from −70 to +50 mV in 10 mV increments. The maximal current densities (pA/pF) were calculated and analyzed.

## Behavioral test

Electronic Von Frey hair algesiometer (IITC Life Science) was used to measure the hind paw withdrawal threshold. Before starting the test, mice were separately placed in elevated Plexiglas chambers on metal mesh flooring for 30 mins. A von Frey hair with bending force (0.6 g, 1 g, 1.4 g, 2 g, 4 g) was exerted perpendicular to the plantar surface of the hind paw until it just bent and the hind paw of mice of elevated. The force displayed on the electronic device were recorded. The threshold force required to elicit withdrawal of the paw was determined three times on each hind paw and averaged.

Gait analysis was performed on mice 4 weeks after ACLT by the CatWalk system (Noldus) according to our previous protocol (*Zhen et al., 2013*). Briefly, each mouse was placed walkway and allowed to allow the free movement from one side to the other side for at least three times. Mice were trained previously in the formal experiment. After the recording of mouse gait, several parameters were generated, and 5 of the most relevant parameters to OA pain were analyzed. (1) stands, (2) maximal contact at. (3) maximal (4) swing and (5) single stance.

## Western blotting and ELISA

Western blotting was performed on the lysates of DRG neuron culture and tibial subchondral bone marrow. The samples were separated by SDS-PAGE gel and transferred onto a nitrocellulose

membrane (Bio-Rad Laboratories). After incubation with specific primary and secondary antibodies, signals were detected by an enhanced chemiluminescence kit (Amersham Biosciences). The primary antibodies used are as follow rabbit anti-Na$_V$1.8 (1:500, ASC-016, Alomone), rabbit anti-CREB (1:2000, #9179, Cell Signaling Technology), rabbit anti-pCREB (1:1000, 9198, Cell Signaling Technology), rabbit anti-PKA c- **c-α** (1:1000, 4782, Cell Signaling Technology) and rabbit anti-GAPDH (1:1000, 5174, Cell Signaling Technology). The experiments were repeated three times and a representative film was selected.

To measure the concentration of PGE2 in the subchondral bone marrow of mice tibiae, a PGE2 ELISA kit (514010, Cayman) was used according to the manufacturer's manual. Briefly, we harvest the subchondral bone and then homogenized by ultrasound. The supernatant was aspirated after high-speed centrifugation (13,200 g) for 10 mins. The concentration of PGE2 was normalized by total protein concentration using the BCA assay.

## ChIP assay

The ChIP assay was carried out using the epiquik ChIP Kit (Epigentek catalog number: P-2002–1). Briefly, the cultured lumbar DRG cells were crosslinked with 1% formaldehyde at for 10 min. After the collection of the cell, the sonication was performed until the DNA was broken into fragments with a mean length of 200 bps – 500 bps. The samples were subjected to immunoprecipitation with 2 mg of rabbit antibodies against pCreb1 (CST, 1:50) for 90 min at room temperature and 10% of the sample for immunoprecipitation was used as an input (a positive control). After purification, the DNA fragments were amplified using qRT-PCR with the primers for Na$_V$1.8 promoter listed in Supplementary Table 2.

## Retrograde tracing

Retrograde tracing was performed at 3-month-old male SD rats (Charles River Laboratories) (300–400 g, n = 6 per group) 2 months after ACLT. According to the previous study (*Ferreira-Gomes et al., 2010*), a 20 mm parapatellar incision was made over the medial side of the left knee. Ipsilateral femoral and tibial subchondral bone were subject to retrograde labeling. We injected 2 μl DiI (Molecular Probes; with 5 mg/ml in N, N dimethylformamide) into the femoral and tibial subchondral bone areas using a Hamilton syringe with a 27-gauge needle. Immediately after injection, bone wax was used to seal the drilling holes to prevent tracer leakage. Animals were euthanized 2 weeks after retrograde injection and the left lumbar DRGs (L3-5) were isolated for fluorescence detection. Twenty sections from each DRG were used for statistical analysis.

## Statistical analysis

Data are presented as means ± standard deviations. Error bars represent standard deviations. We used unpaired or paired two-tailed Student's t-tests for comparisons between two groups, and one-way ANOVA with Bonferroni post hoc test for multiple comparisons, in comparison between three or more groups, two-way ANOVA with Bonferroni post hoc test were used. All data demonstrated a normal distribution and similar variation between groups. For all experiments, p<0.05 was considered to be significant.

# Acknowledgements

The authors thank David D Ginty (Harvard Medical School) for Cox-2 $^{floxed}$ mice and Jenni Weems and Rachel Box in the editorial office at the Department of Orthopaedic Surgery, The Johns Hopkins University for manuscript editing.

# Additional information

## Funding

| Funder | Grant reference number | Author |
|---|---|---|
| National Institutes of Health | AR071432 | Xu Cao |
| National Institutes of Health | AR063943 | Xu Cao |

The funders had no role in study design, data collection and interpretation, or the decision to submit the work for publication.

### Author contributions
Jianxi Zhu, Conceptualization, Resources, Formal analysis, Visualization, Methodology, Writing - original draft, Writing - review and editing; Gehua Zhen, Supervision, Investigation, Visualization; Senbo An, Data curation, Software, Formal analysis; Xiao Wang, Data curation, Software, Investigation, Methodology; Mei Wan, Data curation, Methodology, Project administration; Yusheng Li, Software, Formal analysis, Visualization, Methodology; Zhiyong Chen, Data curation, Formal analysis; Yun Guan, Supervision, Validation, Project administration; Xinzhong Dong, Supervision, Project administration, Writing - review and editing; Yihe Hu, Conceptualization, Supervision, Project administration; Xu Cao, Conceptualization, Funding acquisition, Investigation, Project administration, Writing - review and editing

### Author ORCIDs
Jianxi Zhu (iD) https://orcid.org/0000-0003-4637-0704
Mei Wan (iD) http://orcid.org/0000-0001-9404-540X
Xinzhong Dong (iD) http://orcid.org/0000-0002-9750-7718
Xu Cao (iD) https://orcid.org/0000-0001-8614-6059

### Ethics
Human subjects: human study was approved by the Johns Hopkins Medicine Institutional Review Boards. Written informed consent and consent to publish forms were obtained from all volunteers prior to providing samples. (Protocol number: Mo18M308).
Animal experimentation: All animal experiments were approved by the Institutional Animal Care and Use of Johns Hopkins University, School of Medicine. (Protocol number: Mo18M308).

### Decision letter and Author response
Decision letter https://doi.org/10.7554/eLife.57656.sa1
Author response https://doi.org/10.7554/eLife.57656.sa2

## Additional files

### Supplementary files
• Transparent reporting form

### Data availability
Source data files have been provided for Figures 1–6.

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
