## [Decision Letter]

**Acceptance summary:**

The manuscript provides compelling new evidence to support the conclusion that subchondral bone marrow resorption is related to osteoarthritis (OA) pain. Specifically, the authors nicely link prostaglandin production by osteoblasts to the activation of the sodium channel Na_V_1.8. This is seen as a very active area of investigation with strong clinical relevance.

**Decision letter after peer review:**

Thank you for submitting your article "Aberrant subchondral osteoblastic metabolism modifies Na_V_1.8 for osteoarthritis" for consideration by *eLife*. Your article has been reviewed by three peer reviewers, one of whom is a member of our Board of Reviewing Editors, and the evaluation has been overseen by Clifford Rosen as the Senior Editor. The reviewers have opted to remain anonymous.

The reviewers have discussed the reviews with one another and we have drafted this decision to help you prepare a revised submission.

We would like to draw your attention to changes in our revision policy that we have made in response to COVID-19 (https://elifesciences.org/articles/57162). Specifically, we are asking editors to accept without delay manuscripts, like yours, that they judge can stand as *eLife* papers without additional data, even if they feel that they would make the manuscript stronger. Thus, the revisions requested below only address clarity and presentation.

Summary:

The reviewers believe that the study is well done with appropriate controls and provides a clear conclusion consistent with the data. The implications extend beyond bone pathophysiology into the interaction between nervous system and bone metabolism in general terms and could initiate novel therapeutic approaches. Nevertheless, several issues need to be addressed to further strengthen the manuscript.

Revisions:

1) The authors claim that osteoblast are central to inducing OA pain by secreting excessive PGE2, which is clearly novel and interesting. A question remains whether osteoclasts are also involved in inducing OA pain and, if so, what might be the mechanism?

2) Na_v_1.8 is an important ion channel for chronic pain sensation. Could a Na_v_1.8 inhibitor be effective for OA pain treatment and disease progression? Please expound on this concept.

3) The authors developed a bone targeting conjugate drug for OA disease control based on the aberrant osteoblastic function in OA. How is this conjugate linked to osteoblast metabolism and Na_v_1.8 function? Please describe the drug target and working mechanistic hypothesis.

4) The relationship between OA pain and disease progression remains unclear, as noted by the authors. The anti-NGF antibody tanezumab shows efficacy in OA pain, even while certain patients suffer a rapid progression of joint degeneration after tanezumab therapy. The authors are encouraged to offer plausible explanations regarding this apparent clinical dissociation.

5) The authors stated: "Human genetic mutations of Na_V_1.8 directly induce pain hypersensitivity". The authors should make clear that the mutation identified in the promoter region of the human *Scn10a* gene (encoding for Na_v_1.8) is the gain-of-function mutation.

6) The authors need to justify why they chose ACLT OA model, instead of DMM OA model.

7) In figure legends, the authors need to disclose if one-way or two-way ANOVA was used followed by the Bonferroni post hoc test. Some of the data involving two parameters need to use two-way ANOVA for statistical analysis. For example, Figure 1G involves different time points and different animal models, two-way ANOVA should be used.

8) The authors do need to describe which statistical method was used following ANOVA in the figure legends.

9) The authors need to provide more information on *Pirt-GCaMP3^fl/-^* mice.

10) The authors have used the electronic Von Frey method to test behavior. Is there any evidence to support the choice of electronic Von Frey over manual Von Frey? Please specify if manual Von Frey was also used, and if any adjustments were made to distinguish true pain responses from "touch-on" responses and ambulation.

11) The author developed a small molecule conjugate (Aln-Ly) but did not mention the injection method in the animal study – please address this and expand on how the appropriate regime and dosage was planned since the toxicology of this conjugate is unknown.

12) Please indicate how many graders evaluated the OA score in the Materials and methods section.

---

## [Author Response]

Revisions:1) The authors claim that osteoblast are central to inducing OA pain by secreting excessive PGE2, which is clearly novel and interesting. A question remains whether osteoclasts are also involved in inducing OA pain and, if so, what might be the mechanism?

We highly appreciate the positive comments from the reviewers. We agree that osteoblastic PGE2 plays a central role in inducing chronic OA pain during disease progression. Nevertheless, osteoclast also actively participate in OA progression and chronic hypersensitivity induction. As we know that osteoclasts and osteoblasts cooperate in bone metabolism. Our previous work demonstrated that subchondral osteoclasts induce OA progression through TGF-β pathway^[1]^. Moreover, osteoclasts secret axon guidance molecule Netrin-1 to induce sensory nerve innervation in subchondral bone area to induce OA pain hypersensitivity^[2]^. Collectively, we believe that osteoclast and osteoblast cooperatively participate in OA pain induction during aberrant subchondral bone metabolism through secreting a cascade of biologically active molecules.

2) Na_v_1.8 is an important ion channel for chronic pain sensation. Could a Na_v_1.8 inhibitor be effective for OA pain treatment and disease progression? Please expound on this concept.

Na_v_1.8 is the most important voltage gated sodium channel in mechanical pain sensation. As OA pain is mainly mechanical hypersensitivity, we believe that the effectiveness of Na_v_1.8 inhibitor could be at least in two parts: the short term and long term. In the short term, Na_v_1.8 inhibitors could directly block action potential initiation in peripheral sensory nerve endings in subchondral bone area. In the long term, Na_v_1.8 inhibitors attenuate the chronic hypersensitivity which in turn could tunedown the local inflammatory environment as a feedback. In this sense, Na_v_1.8 inhibitors could be highly effective in OA pain management.

3) The authors developed a bone targeting conjugate drug for OA disease control based on the aberrant osteoblastic function in OA. How is this conjugate linked to osteoblast metabolism and Na_v_1.8 function? Please describe the drug target and working mechanistic hypothesis.

The conjugate consists of two basic elements: the bone targeting alendronate and the TGF-β inhibitor. We link these two elements through hydroxyl group bond which could break as the conjugate reaches the bone surface. Since TGF-β is the upstream of aberrant bone metabolism in OA subchondral bone^[1]^, this conjugate could effectively rebalance the overactivated osteoblastic bone metabolism in subchondral bone area. As overactivated osteoblastic metabolism is blocked in subchondral bone, the local PGE2 level is reduced and Na_v_1.8 modification could be halted. Thus, the alendronate TGF-β conjugate could be therapeutically effective for OA pain management and disease progression by targeting the most upstream signals in bone metabolism.

4) The relationship between OA pain and disease progression remains unclear, as noted by the authors. The anti-NGF antibody tanezumab shows efficacy in OA pain, even while certain patients suffer a rapid progression of joint degeneration after tanezumab therapy. The authors are encouraged to offer plausible explanations regarding this apparent clinical dissociation.We are also highly interested in the side effect of this anti-NGF antibody tanezumab^[3]^. The most up to date clinical outcome of tanezumab shows that certain percentage of the patients suffered a rapid progression of the disease after drug administration. A number of theories have been proposed to explain the incidence of advanced arthropathies in this setting, the most common being that improved analgesia leads to increased loading of a diseased joint, resulting in disease progression. This phenomenon could be similar to neuropathic arthropathy (Charcot joint), which occurs secondary to loss of sensation in an extremity and can result in joint dislocations, fractures and fragmentation. However, there might be another biological explanation. NGF levels are high in inflammatory tissues, and these tissues have a high number of monocytes. NGF is reportedly part of an important regulatory pathway in monocytes and other cell types. Inflammatory stimuli activate Toll-like receptors (TLRs) on monocytes, which increases expression of the NGF receptor TrkA. Monocyte could also increase NGF production, and NGF signaling might abate TLR2-mediated cell death, broadening the beneficial effects ofNGF. In summary, NGF inhibitors can markedly reduce chronic musculoskeletal pain, but enthusiasm for these compounds has been dampened by a small number of cases of advanced arthropathy in OA joints. In this sense, novel drug targets of upstream signals in OA pain pathways are needed for a more comprehensive therapeutic approach.

5) The authors stated: "Human genetic mutations of Na_V_1.8 directly induce pain hypersensitivity". The authors should make clear that the mutation identified in the promoter region of the human Scn10a gene (encoding for Na_v_1.8) is the gain-of-function mutation.

Thanks for the reviewers’ suggestion. We have changed the word “mutation” to “gain of function mutation in the promoter region” in main text.

6) The authors need to justify why they chose ACLT OA model, instead of DMM OA model.

ACLT and DMM are the common surgical OA models and they are applicable in some different scenarios. In ACLT models, disease progression is more rapidly with relatively high pain sensitivity which gives us a good opportunity to investigate pain mechanisms and therapies. Subchondral bone erosion of the posterior tibial plateau (in some cases reaching the growth plate) was demonstrated in histologic photomicrographs from the ACLT with more severe subchondral bone metabolism changes. Meanwhile, DMM model was commonly used in slowly progressing studies like cartilage destructions. In our previous study^[2]^, we compared this two models in OA pain induction and found no significant differences between this two models. As a result, we continue to use ACLT model in this study.

7) In figure legends, the authors need to disclose if one-way or two-way ANOVA was used followed by the Bonferroni post hoc test. Some of the data involving two parameters need to use two-way ANOVA for statistical analysis. For example, Figure 1G involves different time points and different animal models, two-way ANOVA should be used.

Thanks for the reviewer’s comment on statistical analysis. We added the 2-way ANOVA statistical analysis in the Materials and method section.

8) The authors do need to describe which statistical method was used following ANOVA in the figure legends.

Thanks for the advice on statistical methods. We added ANOVA with Bonferroni post hoc test in all the figure legends as the Materials and methods section mentioned.

9) The authors need to provide more information on Pirt-GCaMP3^fl/-^ mice.

*Pirt-GCaMP3^fl/-^* mice was developed in Dr Xinzhong Dong’s lab^[4-6]^. In order to monitor the activity of large populations of DRG neurons in intact live animals, they developed an in vivo imaging technique using Pirt-GCaMP3 mice, in which the genetic-encoded Ca^2+^indicator GCaMP3 is specifically expressed in >95% of all DRG neurons under the control of the Pirt promoter. We added the genetic background and functions of this mice in Materials and methods section.

10) The authors have used the electronic Von Frey method to test behavior. Is there any evidence to support the choice of electronic Von Frey over manual Von Frey? Please specify if manual Von Frey was also used, and if any adjustments were made to distinguish true pain responses from "touch-on" responses and ambulation.

Thanks for the reviewer’s interests on the electronic Von Frey test. We highly agree that considerable variance exits between groups in Von Frey tests. For this reason, we take the average numbers by two independent technicians and repeated the experiment by 3 times by the same standards. True response is more like a rapid withdrawal of the paw instantaneously after stimulation which could be differentiated from touch on and ambulation response. The reasons why we think electronic Von Frey may be more applicable are as follows:

1) The direct force is used, and we do not need to calculate the paw withdrawal threshold.

2) The force is more accurate than manual Von Frey meter.

3) Electronic Von Frey could be easy to adjust the force. But manual Von Frey meters are very easy to be bend and not accurate.

11) The author developed a small molecule conjugate (Aln-Ly) but did not mention the injection method in the animal study – please address this and expand on how the appropriate regime and dosage was planned since the toxicology of this conjugate is unknown.

Thanks for the reviewer’s concern on injection method. For conjugate injections, we used intraperitoneal injection method and 1mg/kg per week dosage according to previous toxicological experiments.

12) Please indicate how many graders evaluated the OA score in the Materials and methods section.

The OARSI scores were evaluated by two independent graders and the averages were taken. We added this information in the Materials and methods section.

References:

1. Zhen GH, Wen CY, Jia XF, Li Y, Crane JL, Mears SC, Askin FB, Frassica FJ, Chang WZ, Yao J, Carrino JA, Cosgarea A, Artemov D, Chen QM, Zhao ZH, Zhou XD, Riley L, Sponseller P, Wan M, Lu WW, Cao X. Inhibition of TGF-β signaling in mesenchymal stem cells of subchondral bone attenuates osteoarthritis. Nat Med 2013, 19(6): 704-+.2. Zhu S, Zhu J, Zhen G, Hu Y, An S, Li Y, Zheng Q, Chen Z, Yang Y, Wan M. Subchondral bone osteoclasts induce sensory innervation and osteoarthritis pain. The Journal of clinical investigation 2019, 129(3).3. Lane NE, Corr M. Osteoarthritis in 2016: Anti-NGF treatments for pain—two steps forward, one step back? Nature Reviews Rheumatology 2017, 13(2): 76.4. Miller RE, Kim YS, Tran PB, Ishihara S, Dong XZ, Miller RJ, Malfait AM. Visualization of Peripheral Neuron Sensitization in a Surgical Mouse Model of Osteoarthritis by in vivo Calcium Imaging. Arthritis & Rheumatology 2018, 70(1): 88-97.5. Tang ZX, Kim A, Masuch T, Park K, Weng HJ, Wetzel C, Dong XZ. Pirt functions as an endogenous regulator of TRPM8. Nature communications 2013, 4.6. Patel KN, Liu Q, Meeker S, Undem BJ, Dong XZ. Pirt, a TRPV1 Modulator, Is Required for Histamine-Dependent and -Independent Itch. PLoS One 2011, 6(5).